# Detecting and Characterizing Simulated Sea Breezes Over the U.S. Northeast Coast with Implication for Offshore Wind Energy

Geng Xia[1], Caroline Draxl[1,2], Michael Optis[3], Stephanie Redfern[1]

[1]National Renewable Energy Laboratory, Golden, CO 80401 USA

[2] Renewable and Sustainable Energy Institute, Boulder, CO 80309, USA

[3]Electricity Generation and Development Renewables, San Diego, CA, USA

*Correspondence to*: Geng Xia (Geng.Xia@nrel.gov)

**Abstract.** With the vastly planned offshore wind farm construction along the U.S. East coast, identifying and understanding key coastal processes, such as sea breezes, has become a critical need for the sustainability and development of the U.S. offshore wind energy. In this study, a new two-step identification method is proposed to detect and characterize three types of sea breezes (pure, corkscrew and backdoor) over the U.S. Northeast coast from a year-long WRF (Weather Research and Forecasting) simulation. The results suggest that the proposed detection method can identify the three different types of sea breezes in the model simulation. Key sea breeze features, such as the calm zone associated with pure sea breezes and coastal jets associated with corkscrew sea breezes, are evident in the sea breeze composite imagery. In addition, the simulated sea breeze events indicate a seasonal transition from pure to corkscrew sea breeze between March and August as the land-sea thermal contrast increases. Furthermore, the location and extension of the sea breeze front are different for each type of sea breeze, suggesting that the coastal impact of sea breeze varies with sea breeze type. From the wind energy perspective, the power production associated with a 10 megawatts offshore wind turbine would be approximately 3 to 4 larger during a corkscrew sea breeze event than the other two types of sea breezes. This highlights the importance of identifying the correct type of sea breeze in numerical weather/wind energy forecasting.

## 1. Introduction

Offshore wind began to be harnessed as a renewable energy source in the early 1990's, and has been experiencing a rapid growth ever since due to its high wind resource potential and virtually unlimited installation areas over the ocean (Boyle. 2004; Esteban et al., 2011; Costoya et al. 2020). Europe has a long history of development and investment in offshore wind energy and is expected to have a capacity of 74 GW offshore wind power installed by 2030 (van Hoof et al. 2017). However, despite having a vast offshore wind resource that can reach up to 2000 GW of potential production, the United States currently only has a single 30-MW commercial wind farm installed (Schwartz et al. 2010, Costoya et al. 2020; Musial et al., 2016; Draxl et al. 2015; Deepwater Wind. 2016). According to the U.S. Department of Energy (U.S. DOE), more than 25 offshore wind projects with a total generating capacity of 25 GW are being planned, mostly concentrated along the U.S. East Coast. As many offshore wind farms will be constructed in the foreseeable future, it is essential to identify the meteorological research needs that are critical for the sustainability and growth of U.S. offshore wind energy.

Both Archer et al. 2014 and the Offshore Wind Workshop from DOE (DOE. 2019) addressed this issue and pointed out an essential need to accurately capture the dynamic coastal processes from both observational and modeling perspectives as they represent a significant source of the uncertainty in the offshore wind resource assessment. One of such costal processes is the sea breeze, which is defined as a local circulation induced by a thermal contrast between the land and sea (Simpson, 1994; Miller et al. 2003). The basic structural, dynamic and physical properties of the sea breeze are well documented (Abbs and Physick. 1992; Miller et al. 2003; Crosman and Horel. 2010). Previous studies have mostly investigated the impacts of sea breeze on air quality, heat waves and even flooding events (Yerramilli et al., 2009; Bianco et al., 2006; Golding et al., 2005, Lombardo et al., 2016). Recently, more attention has been given to classifying the different types of sea breezes (Miller et al. 2003; Steele et al. 2013; 2015) based on the orientation of the prevailing wind (PW) with respect to the coastline.

During offshore PW conditions, sea breezes can be classified into three categories: pure, corkscrew and backdoor (Miller et al. 2003). A pure sea breeze occurs when there is no alongshore wind component, whereas a corkscrew (backdoor) sea breeze occurs when the PW has an alongshore component with land to the left (right). There have only been a few studies focusing on onshore PW as atmospheric conditions normally suppress the development of sea-breeze circulations during such cases (Hu and Xue. 2016; Arritt 1993; Finkele et al. 1995). With a large human population living in major cities in proximity to a coastline, as well as the growing development of offshore wind energy, detecting and forecasting sea breeze events is of high importance. Past studies have mostly investigated sea breeze detection methods from observational perspectives (Azorin-Molina et al., 2011; Prtenjak and Grisogono et al. 2007; Laird et al. 2001; Gustavsson et al. 1995; Ryznar et al. 1981). To our best knowledge, Steele et al. 2015 is the only study that proposed a sea breeze detection algorithm from a modeling perspective. Using 11-year model simulations, they presented an identification method that

distinguished sea breeze types along five different coastlines around the southern North Sea of England. Given the upcoming investment for large offshore wind projects along the U.S. East Coast, it is important to develop a numerical detection method for sea breezes over this region as sea breezes constitute an important coastal wind climate and will play an important role in resource assessment during the pre-construction phase.

The objective of this study is to present a new two-step identification method to detect sea breezes using numerical weather simulations. The method is tested along the U.S. Northeastern seaboard, centered on the New York metropolitan area which is subject for major offshore wind farm construction (Redfern et al. 2021), using a year-long Weather Research and Forecasting (WRF) model simulation. Section 2 describes the simulation setup as well as the sea breeze detection method. The results from the detected sea breeze events using the proposed method are analyzed in detail in Section 3. Section 4 examines the uncertainty associated with the method and discusses the offshore wind potential with respect to each type of sea breezes. followed by the conclusion in Section 5.

## 2. Experiment design and sea breeze identification method

### 2.1 Experiment design

The WRF model version 4.12 (Skamarock et al. 2008; Powers et al. 2017) is used to conduct the model simulation in this study. Table 1 summarizes the main physical options used (Pronk et al. 2021). Both the boundary and initial conditions are initialized with the ERA5 dataset (Hersbach et al., 2020). The initial sea surface temperature field is updated by the Operational Sea Surface Temperature and Sea Ice Analysis product (Stark et al., 2007). Static fields, such as topography height, land use category, soil and vegetation, use the default options from the WRF Prepossessing System.

**Table 1**: Summary of WRF Model Configuration

| | |
|---|---|
| Shortwave | Rapid Radiative Transfer Model for GCMs (Iacono et al., 2008) |
| Longwave | Rapid Radiative Transfer Model for GCMs (Iacono et al., 2008) |
| Microphysics | Eta Grid-scale Cloud and Precipitation (Tao et al. 2003) |
| Boundary Layer | Mellor-Yamada-Nakanishi-Niino (MYNN; Nakanishi and Niino, 2006, 2009) |
| Land Surface | Noah Land Surface Model (Chen and Duhia., 2001) |
| Surface Layer | MYNN |
| Cumulus | Kain-Fritsch (Kain and Fritsch., 1990; Kain., 2004) |

The model has a two-way interactively nested domains with horizontal grid spacings of 6 km and 2 km respectively (Fig.1a). The domains cover the U.S. Northeastern seaboard and near-coastal Atlantic Ocean. Atmospheric nudging is applied on the outer domain every 6 hours. There are 61 vertical levels and the lowest 9 levels are within 200 meters (m) above the ground. The 1-year simulations span from September 2019 to September 2020 and are conducted using a monthly reinitialization

method with a 2-day spin-up and 1-day extension for each monthly simulation. These three days are discarded, and the rest
are kept for the analysis. This period is chosen as it aligns with offshore floating lidar observations for validation of the WRF
model runs (Optis et al. 2020)

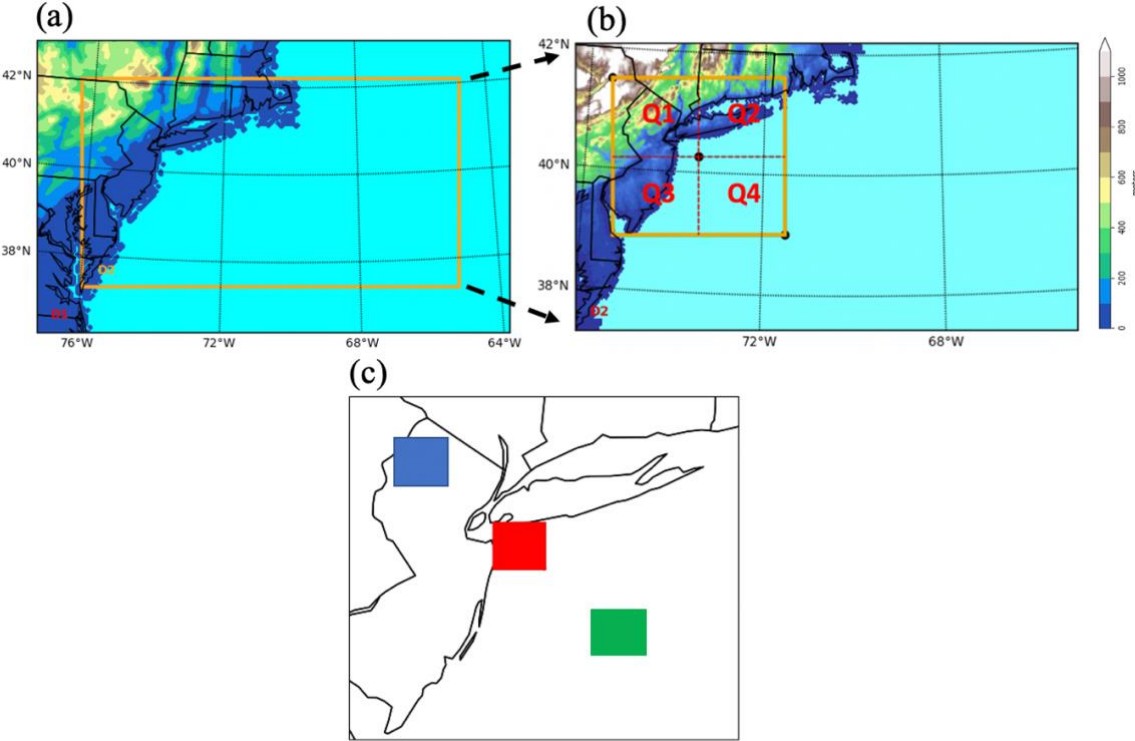

**Figure 1:** Spatial  map for domain 1 (a), domain 2 (b) and the New York metropolitan area (c). The brown box in domain 2
marks the New York metropolitan area, and Q1, Q2, Q3 and Q4 represent quadrant 1, quadrant 2, quadrant 3 and quadrant 4
respectively. The blue, red and green boxes in c) mark the regions over land, coast, and ocean which are subject for
variability analysis.

### 2.2 Sea breeze identification method

There are various sea breeze identification methods based on different approaches, such as using satellite remote sensing
(Damato et al..2003), a sea-breeze index (Biggs and Graves 1962; Lyons 1972) and conditional filtering (Azorin-Molina et
al., 2011; Steele et al. 2015). Even though each approach differs in terms of sophistications, all methods ultimately determine
the likelihood of sea breezes based on key meteorological principles (e.g. land-sea temperature contrast) and preceding/co-
existing physical conditions associated with sea breezes (e.g. large-scale wind field or changes in wind speed/wind
directions). Since the accuracy of the detection algorithm strongly depends on the features of the study area (e.g., geometry

of the coastline), data and testing criteria, the filtering approach is selected as the preferred method for this study as it allows for the systematic determinations of days when conditions favor sea breeze formation over a wide range of scenarios.

To automatically select sea breezes event from the model simulation, key meteorological variables that are sensitive to sea breeze existence must be evaluated. Therefore, hourly model output of 10-m wind speed (WS10), wind direction (WD10), 2-m air temperature (T2) as well as sea level pressure (SLP) are analyzed. The search for sea breeze events is temporally constrained to the hours between 08 to 20 local time (LT), as land breezes are more likely to occur at night.

Figure 2 shows the flow diagram of the two-step approach to identify sea breeze events from the model simulation. The first step is applied at a regional scale to identify days with flow regimes that have potential for pure, backdoor and corkscrew sea breeze development. Figure 1b shows the topography map for domain 2 in which the New York metropolitan area is highlighted as our targeted region; that domain is further separated into four quadrants. Quadrant 1 (Q1) is mostly over land while Q4 is over the ocean. Both Q2 and Q3 are over the coastal region. The four quadrants are designed in such a way that the mean meteorological conditions can be representative of those from land (Q1), ocean (Q4) and coastal region (Q2 and

Q3).

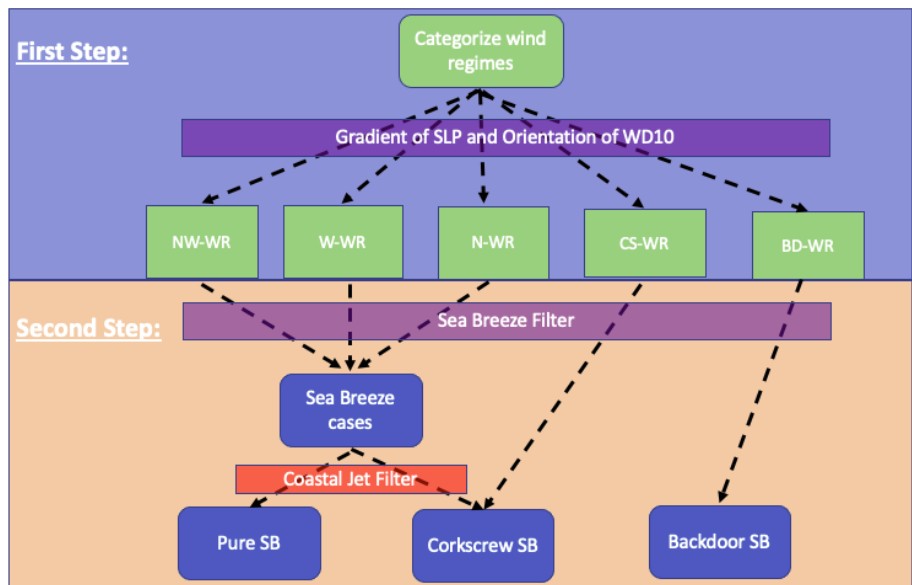

**Figure 2**. Flow diagram showing the filters involved in the sea-breeze identification method for the 1-year (Sept. 2019-Aug. 2020) simulation period. SLP, WD10, SB and WR stand for sea level pressure, wind direction at 10-meter, sea breeze and wind regime respectively.


To facilitate wind regime classification, three idealized SLP conditions for offshore PW are defined (Figure 3), in which the direction of the PW is determined by the balance between pressure gradient force and Coriolis force. Due to the geometry of the coastline, SLP conditions from Figure 3a) are favorable for the development of pure sea breezes, whereas those depicted in Figures 3b) and 3c) favor backdoor sea breezes and corkscrew sea breezes respectively. Further, we have classified the offshore PW into five different wind regimes: i) northwesterly wind regime (NW-WR), ii) westerly wind regime (W-WR), iii) northerly wind regime (N-WR), iv) backdoor sea breeze wind regime (BD-WR) and v) corkscrew sea breeze wind regime (CS-WR). Note that the first three wind regimes all fall into the pure sea breeze wind regime category.

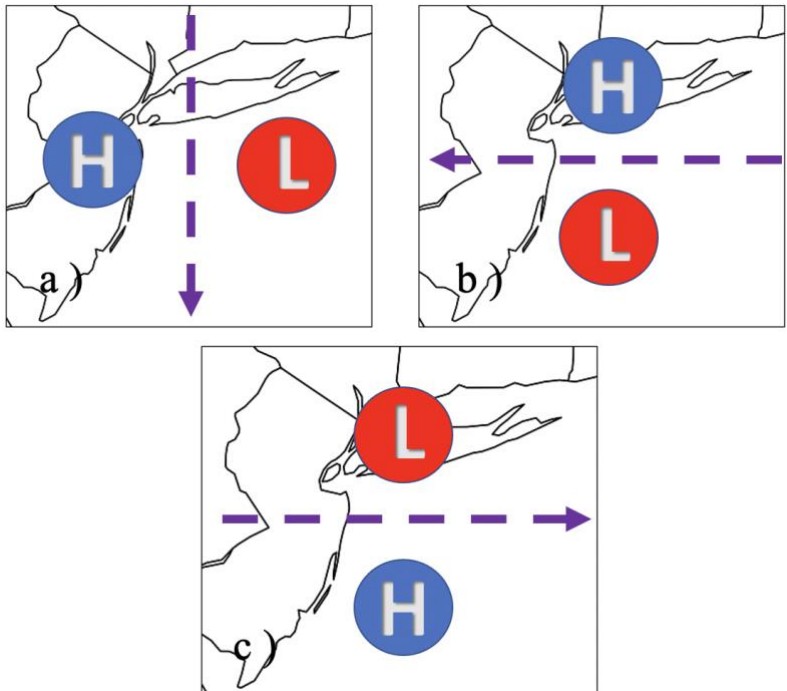

**Figure 3:** Schematics of the three idealized sea level pressure (SLP) conditions and their associated prevailing wind direction over the targeted region. The blue circle represents high pressure and the red circle represents low pressure. The purple arrow indicates the direction of the prevailing wind where SLP conditions favor the development of: a) pure sea breeze; b) backdoor sea breeze; c) corkscrew sea breeze.

For each day at 08 LT, the mean SLP, WS10, and WD10 are calculated in all four quadrants. Days with cyclonic conditions over the targeted region are rejected, as sea breeze identification would be difficult due to rapid changes in wind direction in these cases. Then, the mean SLP and WD10 for each individual quadrants, as well as those for the entire targeted region, are used to determine which wind regime is dominant for that particular day. Table 2 shows the total number of days identified for each wind regime over the course of the simulation. The NW-WR occurs the most often and N-WR is the least common. The other three wind regimes have almost an equal amount of identified days. Figure 4 shows the spatial pattern of averaged

SLP, WS10 and WD10 for all five different wind regimes at 08 LT. Evidently, each wind regime is characterized with a unique SLP condition in term of spatial pattern and magnitude. All the identified days have the potential for sea-breeze development and are further examined in the next step. The unidentified days are rejected as they tend to have a strong onshore wind component and synoptic setups that in general suppress sea-breeze development (Hu and Xue. 2016).

**Table 2**: Number of days for each wind regime from the one-year WRF simulations

| Wind regime | Number of days |
|---|---|
| Northwesterly | 67 |
| Westerly | 51 |
| Northerly | 30 |
| Backdoor | 47 |
| Corkscrew | 51 |
| Unclassified | 120 |

The second step applies the conditional filters to detect sea breeze existence for the days identified from the first step. The following three empirical criteria, which are based on the most important meteorological characteristics associated with sea breeze development, are used. If all the criteria are satisfied, then it is assumed that a sea breeze forms on that particular day.

1) The daytime maximum land-sea temperature difference is greater than 1.5 K
2) The mean mid-day (8am to 3pm) offshore WS10 is smaller than 6.0 m/s.
3) Changes in the mean WD10 for at least two of the four quadrants pass a threshold (75 for NW-WR, 50 for W-WR, 90 for N-WR, 45 for BD-WR and 25 for CS-WR)

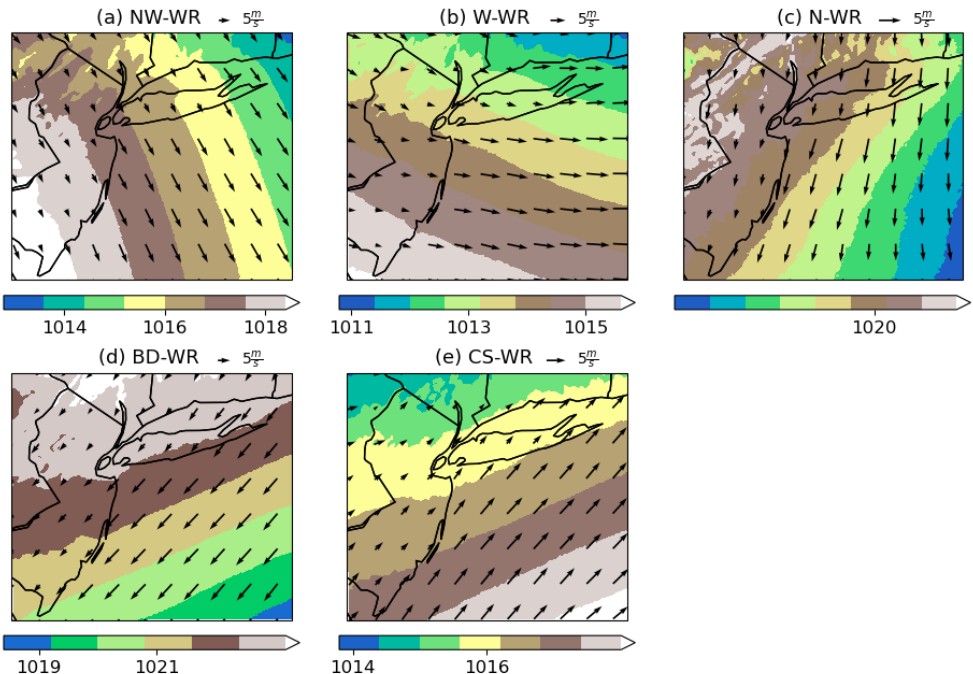

**Figure 4:** Spatial patterns of the averaged sea level pressure (mb), 10-m wind speed (m/s) and wind direction for the five different wind regimes at 08 local time: a) composite average for the northwesterly wind regime (NW-WR); b) composite average for the westerly wind regime; c) composite average for the northerly wind regime (N-WR); d) composite average for the backdoor sea breeze wind regime (BD-WR); e) composite average for the corkscrew sea breeze wind regime (CS-WR).

The first filter examines the thermal contrast between land and sea, as this is the most important physical condition needed for sea breeze formation. The land-sea temperature difference is determined by the mean T2 different between Q1 and Q4. In principle, a positive land-sea temperature difference ($\Delta T > 0°C$) is required for the sea breeze formation. Previous studies have selected different thresholds for $\Delta T$ (Azorin-Molina et al., 2011; Steele et al. 2013, 2015). In this study, the threshold for $\Delta T$ is calculated from March 2020, which generally marks the beginning of the warm season. The second filter aims to eliminate days when the magnitude of offshore PW during the daytime is too strong, as studies (Miller et al. 2003, Steele et al. 2013) have shown that this condition would prevent a sea breeze from reaching inland. Therefore, the mean WS10 from Q1 is calculated and applied in this filter. The critical value of 6 m/s is selected following the work from Steele et al. (2013). The third filter examines the changes in WD10 over the entire region as a shift in wind direction from offshore to onshore is a key characteristic accompanying the sea breeze formation. Due to the difference in wind regimes, the threshold for the change in WD10 varies by sea breeze type to ensure onshore wind flow.

If all the three criteria are fulfilled for the BD-WR and SC-WR cases, these are then automatically identified as backdoor sea breezes and corkscrew sea breezes with no further steps needed. Those falling within the other three wind regimes, however, still need one more filter to distinguish between the corkscrew sea breeze and the pure sea breeze. This filter examines whether WS10 near the coast has increased significantly (by 4.0 m.s$^{-1}$) in the late afternoon (18 to 20 LT) compared with that in the early morning (08 to 10 LT). The point of reference for this filter is at the center of the targeted region (Figure 1b), which is about 20 km away from the coastline. If this condition is satisfied, the identified event will be considered a corkscrew sea breeze. Otherwise, the event is classified as a pure sea breeze. The uncertainties associated with the selected thresholds are discussed in detail in Section 4.

To further quantify the variability of the identified sea breeze cases for each sea breeze type, three areas which are located on land (blue), over the coast (green) and ocean (green) are defined (Figure 1c). The size of each area is about 3 % of the targeted region. For each sea breeze type, the standard deviation of WS10 and WD10 from the identified sea breeze events are calculated to understand how much they differ from the mean state.

## 3. Results

### 3. 1 Temporal variability of the sea breeze events from the model simulation

Using the proposed identification method, a total number of 61 sea breeze events have been identified from the year-long WRF simulation. This includes 28 pure sea breeze events, 24 corkscrew sea breeze events and 9 backdoor sea breeze events. Figure 5 shows the monthly numbers of each type of sea breeze events from September 2019 to August 2020. Overall, the season length for sea breeze occurrence is from February through October. Due to the negative land-sea thermal contrast, sea breezes rarely occur during the cold season.

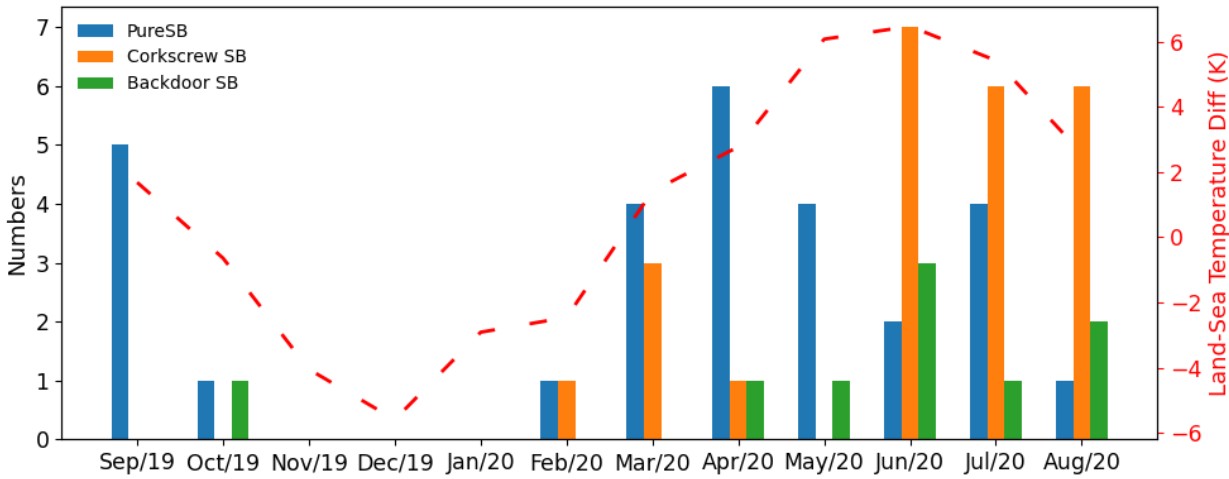


**Figure 5:** Monthly timeseries of identified sea breeze events from September 2019 to August 2020 showing pure sea breezes (blue), the corkscrew sea breezes (orange) and the backdoor sea breezes (green). The red dashed line indicates the monthly averaged land-sea temperature difference at 2 meters.

Out of the three sea breeze types, the backdoor sea breeze is the least common of the three types, which aligns with the findings of Steele et al. (2015). In addition, they are quite evenly distributed from April to August. The most frequent sea breeze types are the pure sea breeze and the corkscrew sea breeze. Note that the total numbers of these two types of sea breeze are relatively even. However, there is a significant difference between the two in terms of seasonal distribution. In general, the pure sea breezes mostly occur during the spring and fall season whereas the corkscrew sea breezes appear more

frequently during the summer months. This suggests that the modeled sea breezes show a gradual change from pure sea breeze to corkscrew sea breeze during the warm season. This could be partially associated with the increase in the land-sea thermal contrast. As the land-sea temperature difference becomes more strongly positive, there is greater potential for corkscrew sea breeze development over pure sea breeze development along the U.S. Northeast coast.

**3.2 Spatial variability of the sea breeze composites**

Figure 6 shows the spatial pattern of the development of the averaged pure sea breeze events from 08 LT to 20 LT. At 08 LT, the simulated WD10 is mostly northwesterly with weaker wind speed over land than that over the ocean. The calm zone, which is the key precursor for pure sea breeze, forms at around 10 LT between 0 to 20 km offshore, reaches its mature stage at around noon and then moves away from the coastline. In conjunction with the development of the calm zone, the near

coastal wind direction shifts from offshore (northwesterly) to onshore (southerly) and the associated WS10 firstly decreases and then increases, reaching a maximum wind speed about 7 to 8 m/s during the late afternoon. Even though the composite wind speed over the calm zone is between 2 and 4 m/s, it falls primarily between 0 and 1 m/s for each individual case. In addition, the location of the calm zone varies by cases, although most calm zones develop relatively close to the coastline.

Table 3 shows the standard deviation of WS10 and WD10 from the simulated pure sea breeze events from 08 LT to 20 LT. The result indicates that the variability of WS10 is largest during the morning hours and decreases after that. Overall, the variability of WS10 is greater over the ocean than that on land. As for WD10, the variability is large during the morning hours because the pure sea breeze is identified from potential days of three different wind regimes (Northwesterly, Northly and Westly) due to the complex shape of the coastline. In addition, variability of WD10 drastically decreases after the

morning hour due to the influence of sea breeze development. Note that the standard deviation of WD10 over the ocean is relatively large until late afternoon. This is mainly due to the development of the calm zone (Figure 6). After the calm zone moves away from the coast, standard deviation of WD10 reduces significantly (16 LT to 20 LT).

Table3 : Standard deviation of WS10 and WD10 of the simulated pure sea breeze cases

| Standard Deviation of WS10(m/s) | | | | | | | | | | | | |
|---|---|---|---|---|---|---|---|---|---|---|---|---|
| | 08 LT | 09 LT | 10 LT | 11 LT | 12 LT | 13 LT | 14 LT | 15 LT | 16 LT | 17 LT | 18 LT | 19 LT | 20 LT |
| Inland | 1.4 | 1.3 | 1.2 | 1.1 | 1.2 | 1.3 | 1.4 | 1.3 | 1.1 | 0.9 | 1.0 | 0.8 | 0.8 |
| Coast | 2.3 | 2.0 | 1.8 | 1.6 | 1.5 | 1.4 | 1.3 | 1.4 | 1.4 | 1.4 | 1.3 | 1.2 | 1.4 |
| Ocean | 3.0 | 2.8 | 2.5 | 2.1 | 1.8 | 1.7 | 1.7 | 1.6 | 1.5 | 1.6 | 1.7 | 1.8 | 1.8 |
| Standard Deviation of WD10 (°) | | | | | | | | | | | | |
| | 08 LT | 09 LT | 10 LT | 11 LT | 12 LT | 13 LT | 14 LT | 15 LT | 16 LT | 17 LT | 18 LT | 19 LT | 20 LT |
| Inland | 119 | 111 | 94 | 77 | 62 | 58 | 58 | 62 | 63 | 63 | 56 | 51 | 55 |
| Coast | 102 | 119 | 114 | 105 | 92 | 78 | 65 | 60 | 55 | 54 | 53 | 62 | 51 |
| Ocean | 118 | 129 | 112 | 108 | 107 | 116 | 115 | 110 | 106 | 97 | 85 | 78 | 76 |


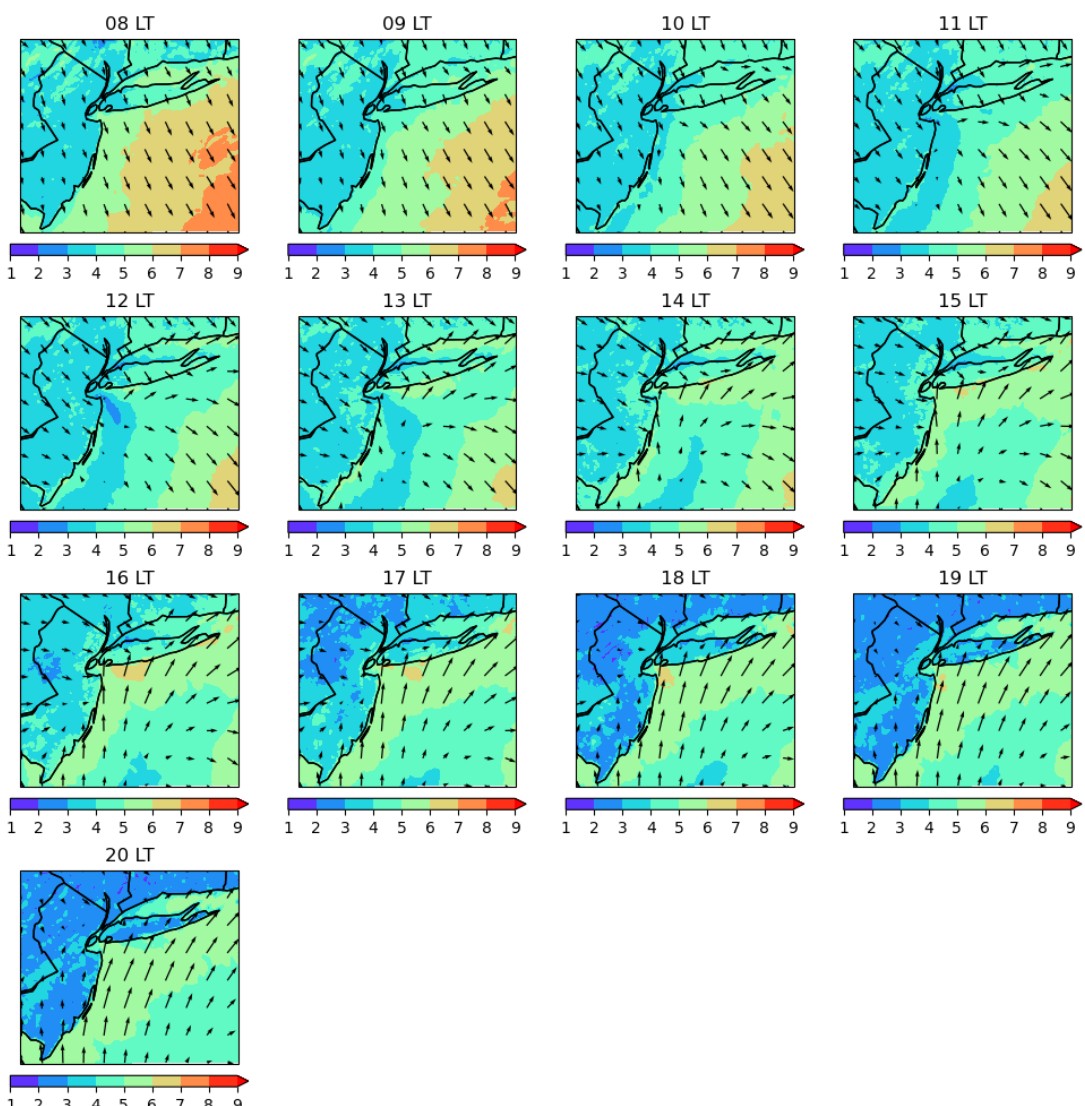

**Figure 6:** Spatial patterns of the development of the averaged pure sea breeze events from 08 local time (LT) to 20 LT. The colored contours indicate the magnitude of 10-m wind speed (m/s) and the vectors show the wind direction.


Figure 7 is similar to Figure 6 but shows the development of averaged corkscrew sea breeze events. At 08 LT, the inland WD10 is mostly westerly, whereas it is southwesterly over the ocean. The arrival of the corkscrew sea breeze is marked by a gradual backing (rotation in the counterclockwise direction) of the wind. The coastal jet, which is the most distinctive feature associated with corkscrew sea breeze, starts to develop around noon time and becomes apparent in the late afternoon.

Previous studies have indicated that the potential causes for a coastal jet are Coriolis acceleration acting on a shore-parallel flow (Hunt et al. 2004) and the presence of a significant topographic barrier (Moore and Renfrew. 2005). The magnitude of

WS10 at the jet core can reach up to 11 m/s, whereas the onshore wind speed falls between 5 and 6 m/s. With regards to offshore wind energy, accurately forecasting the occurrence and timing of the corkscrew sea breeze would be of high value for the wind energy industry. This will be further discussed in the Section 4.


Table 4 is similar to Table 3 but shows the results from the simulated corkscrew sea breezes. In general, the characteristics are similar to that from the pure sea breeze cases. One important aspect is that the variability of WD10 and WS10 over the coastal region are small during the late afternoon hours. This suggests that the position of the simulated jet core (Figure 7) over this region is rather stable among the identified cases, which would have significant offshore wind energy implication

in terms of wind turbine positioning.

Table4 : Standard deviation of WS10 and WD10 of the simulated corkscrew sea breeze cases

| | 08 LT | 09 LT | 10 LT | 11 LT | 12 LT | 13 LT | 14 LT | 15 LT | 16 LT | 17 LT | 18 LT | 19 LT | 20 LT |
|---|---|---|---|---|---|---|---|---|---|---|---|---|---|
| Standard Deviation of WS10(m/s) | | | | | | | | | | | | | |
| Inland | 1.1 | 1.0 | 1.1 | 1.1 | 1.1 | 1.0 | 0.9 | 0.8 | 0.6 | 0.8 | 1.0 | 0.6 | 0.7 |
| Coast | 1.8 | 1.7 | 1.6 | 1.7 | 1.9 | 1.7 | 1.8 | 1.7 | 1.8 | 1.8 | 1.7 | 1.6 | 1.8 |
| Ocean | 2.4 | 2.3 | 2.1 | 1.9 | 1.8 | 1.9 | 2.1 | 2.2 | 2.3 | 2.5 | 2.4 | 2.4 | 2.2 |
| Standard Deviation of WD10 (°) | | | | | | | | | | | | | |
| Inland | 68 | 67 | 58 | 48 | 42 | 43 | 44 | 43 | 43 | 44 | 40 | 46 | 40 |
| Coast | 79 | 80 | 69 | 55 | 45 | 36 | 32 | 35 | 28 | 28 | 28 | 35 | 31 |
| Ocean | 67 | 78 | 85 | 75 | 75 | 78 | 66 | 62 | 60 | 57 | 48 | 36 | 31 |


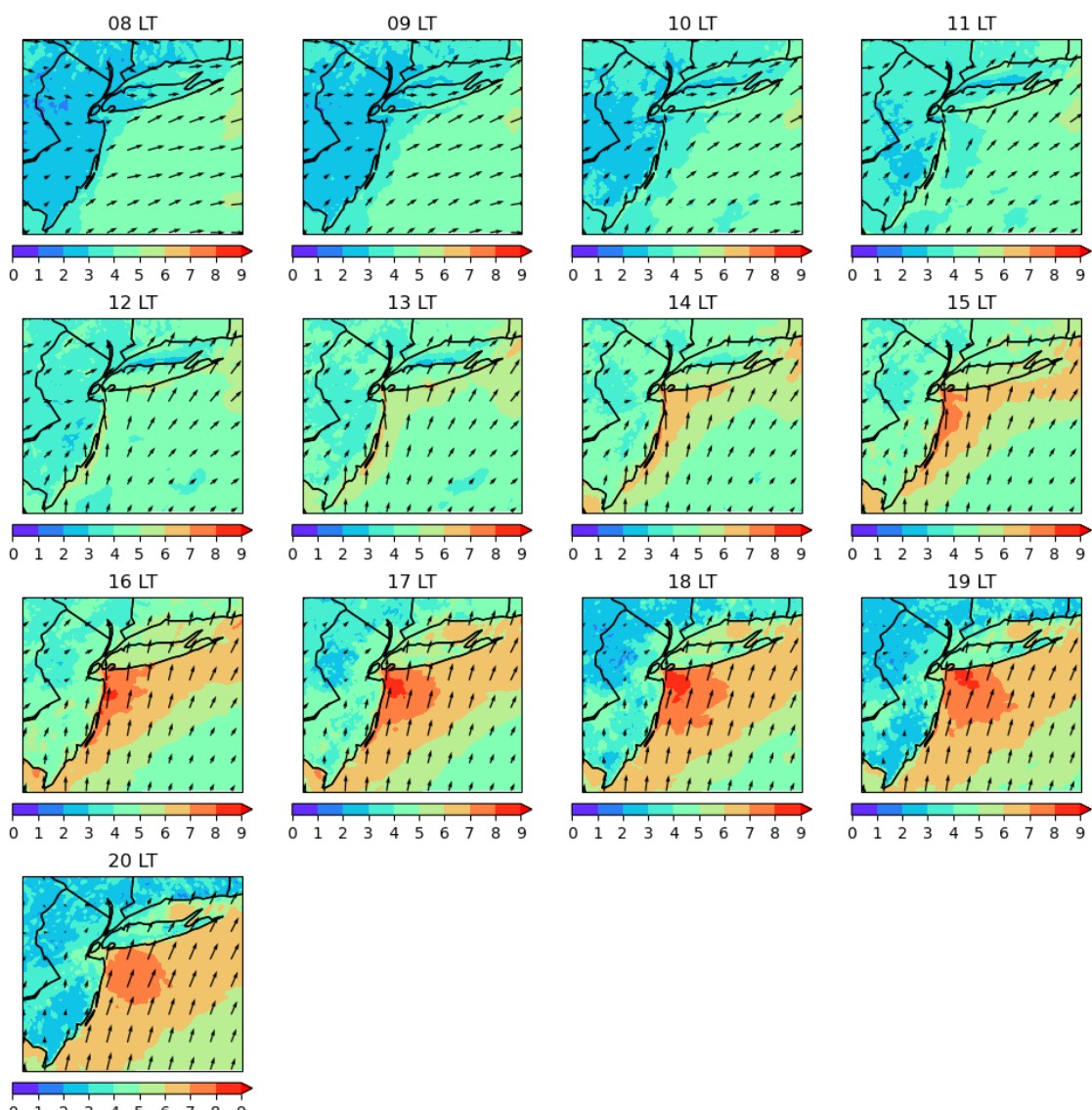

**Figure 7:** Similar to Figure 5, but for the averaged corkscrew sea breeze events

Figure 8 shows the development of averaged backdoor sea breeze events. During the morning hours, the WD10 is mostly northeasterly. In addition, the WS10 over the ocean is significantly faster than those during the pure and corkscrew sea breezes due to less inland deacceleration. In contrast to the corkscrew sea breeze, the arrival of the backdoor sea breeze is marked by a gradual veering (rotation in the clockwise direction) of the wind direction. Additionally, a relatively weak calm zone begins to appear near the coast around noon time as the wind shifts onshore. These features continue to persist 270 throughout the rest of the analysis period. Compared with the other two sea breeze types, the backdoor sea breeze is the least

common sea breeze type over the study region and has the weakest coastal wind speed later in the day. In addition, the variability of WS10 and WD10 are the smallest compared to the other two types of sea breeze (Table 5), which also suggestes that the development of the individual backdoor sea breeze does not differ much from the mean condition (Figure 8).

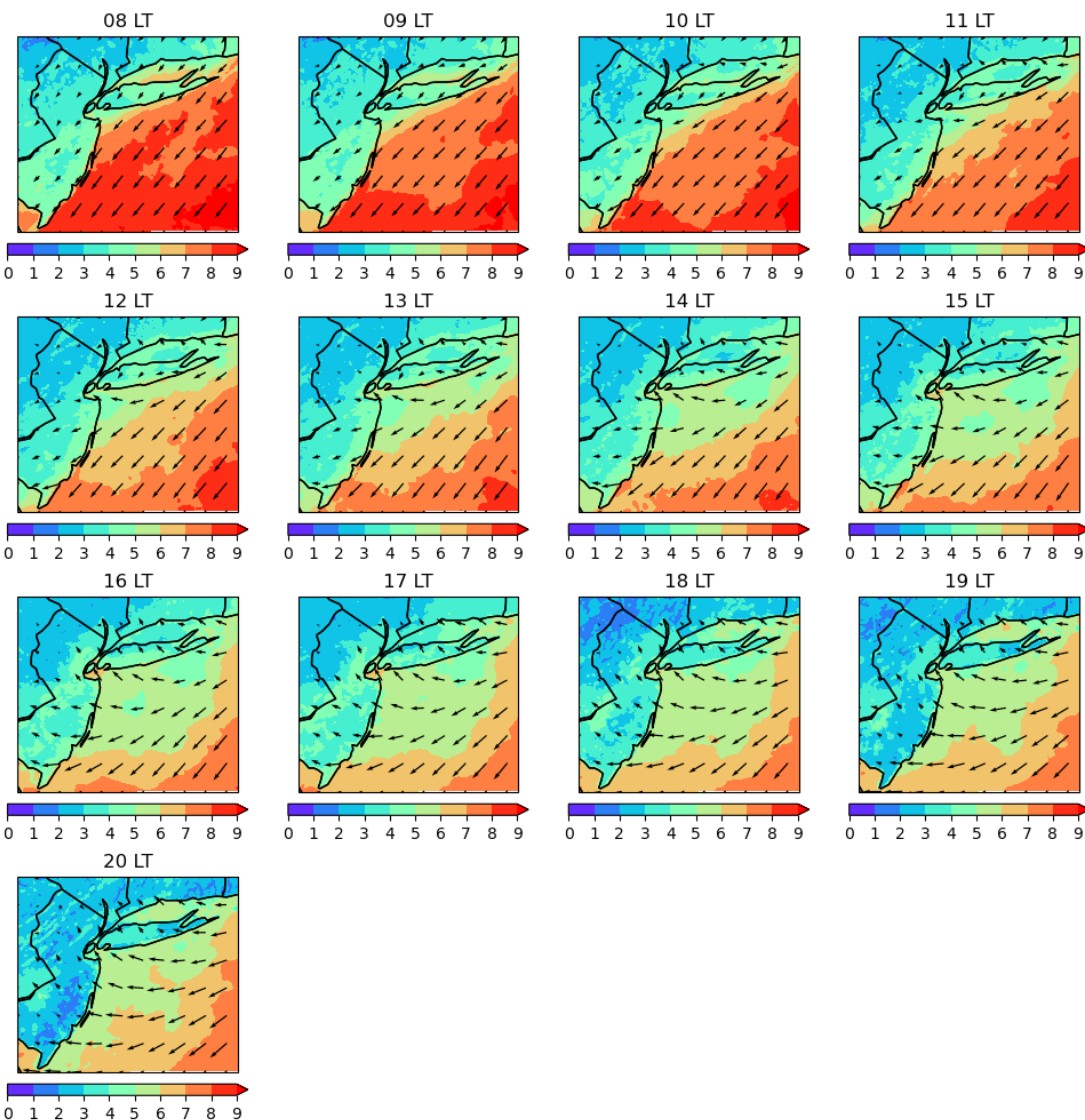


**Figure 8**: Similar to Figure 5, but for the averaged backdoor sea breeze events


Table5 : Standard deviation of WS10 and WD10 of the simulated backdoor sea breeze cases

| Standard Deviation of WS10 (m/s) | | | | | | | | | | | | |
|---|---|---|---|---|---|---|---|---|---|---|---|---|
| | 08 LT | 09 LT | 10 LT | 11 LT | 12 LT | 13 LT | 14 LT | 15 LT | 16 LT | 17 LT | 18 LT | 19 LT | 20 LT |
| Inland | 0.9 | 1.1 | 1.6 | 1.7 | 1.6 | 1.7 | 1.7 | 1.7 | 1.8 | 1.4 | 0.8 | 0.4 | 0.3 |
| Coast | 2.4 | 2.3 | 2.2 | 2.4 | 2.5 | 2.4 | 2.3 | 2.3 | 2.3 | 2.3 | 2.2 | 2.3 | 2.4 |
| Ocean | 3.5 | 3.7 | 3.6 | 3.5 | 3.4 | 3.0 | 2.9 | 3.0 | 3.0 | 3.0 | 2.9 | 2.8 | 2.7 |

| Standard Deviation of WD10 (°) | | | | | | | | | | | | |
|---|---|---|---|---|---|---|---|---|---|---|---|---|
| | 08 LT | 09 LT | 10 LT | 11 LT | 12 LT | 13 LT | 14 LT | 15 LT | 16 LT | 17 LT | 18 LT | 19 LT | 20 LT |
| Inland | 16 | 17 | 20 | 29 | 34 | 37 | 35 | 31 | 31 | 27 | 27 | 29 | 31 |
| Coast | 13 | 14 | 17 | 18 | 16 | 16 | 17 | 16 | 18 | 22 | 22 | 28 | 26 |
| Ocean | 107 | 94 | 42 | 37 | 58 | 65 | 78 | 62 | 36 | 24 | 17 | 22 | 29 |


### 3.3 Coastal impact from the modeled sea breeze events

To estimate the impact of the three types of sea breezes on the coastal region, the 10-m surface divergence is calculated for all the identified sea breeze events. In general, an inland convergence line would form as the sea breeze circulation reaches onshore. Such a convergence line is indicative of a sea breeze front which is the landward edge of the sea breeze system and

is associated with sharp changes in temperature, moisture and wind. Figure 9 shows the spatial map of 10-m divergence from the composite analysis for the three types of sea breezes at 09, 13 and 17 LT. The selected three hours represent the development stage for each type of sea breeze. From the pure sea breeze composite, a significant sea breeze front appears along the coast of New Jersey, Long Island and Connecticut. As for the corkscrew sea breeze, the sea breeze front is strongest along the coast of New Jersey but is much weaker elsewhere. Regarding the backdoor sea breeze, there is a strong

sea breeze front at the center of Long Island and a weaker front along the coast of Connecticut. However, no sea breeze front forms at New Jersey. Furthermore, the inland propagation of the sea breeze front is the greatest for the backdoor sea breeze compared to the other two types.  These results suggest that there is a significant difference in terms of location of sea breeze front with respect to sea breeze type over this region, which further emphasizes the importance of identifying the correct sea breeze type in the numerical weather forecasting.

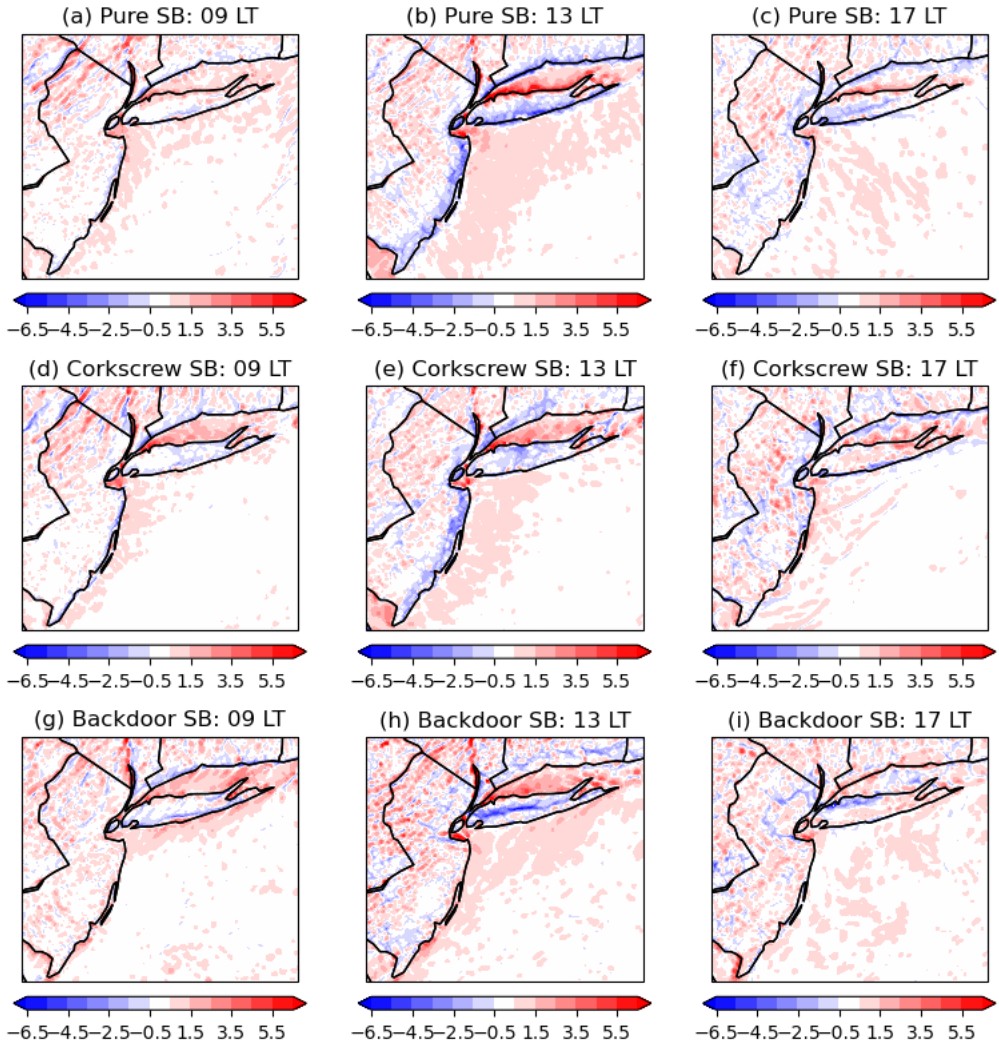

**Figure 9:** Spatial patterns of 10-m divergence ($*10^{-4}$ s$^{-1}$) from the composite average of the pure sea breezes (a-c), corkscrew sea breezes (d-f) and backdoor sea breezes (g-i) at 09, 13 and 17 local time (LT). The positive (negative) value indicates surface divergence (convergence).

## 4. Discussion

The preceding evaluation examines the temporal and spatial variability of the three types of sea breezes. In addition, the possible impact for each type of sea breeze has been discussed. Overall, the results indicate that the proposed sea breeze identification method can detect and characterize sea breeze events from the WRF simulation.

To evaluate the uncertainty of the identified sea breeze events to the chosen criteria, a similar analysis is conducted using first a conservative and then a non-conservative approach. In the conservative (non-conservative) assessment, the threshold for land-sea temperature contrast, midday offshore wind speed, the change in WD10 for each wind regime and the change in WS10 near the coastline is increased (reduced) by 10% of the original value. Table 3 shows the number of detected sea breeze events using the original, conservative and non-conservative approach. Overall, there is a 10 % change in the number

of identified backdoor sea breeze and pure sea breeze events, whereas a greater degree of sensitivity (~25 %) is found for the corkscrew sea breezes. Further analysis indicates that the increase in sensitivity in the case of the corkscrew sea breezes is more associated with the threshold selection for the coastal jet filter. This indicates that more offshore wind observations are needed to determine a more robust threshold value. Figure 10 shows the monthly timeseries of sea breeze events using the conservative and non-conservative approach. In general, the distribution pattern of sea breezes using both criteria look very

similar to that from Figure 5, suggesting consistency in the transition from pure sea breezes to corkscrew sea breezes during the warm season. The spatial composites (figures not shown) show that the key sea breeze features, such as the calm zone for pure sea breezes and coastal jet for corkscrew sea breezes, remain evident.

**Table 3**: Number of identified events for each sea breeze type using conservative, original and non-conservative selection
criteria

|  | Conservative | Original | Non-conservative |
|---|---|---|---|
| **Pure sea breeze** | 25 | 28 | 31 |
| **Corkscrew sea breeze** | 19 | 24 | 31 |
| **Backdoor sea breeze** | 8 | 9 | 10 |

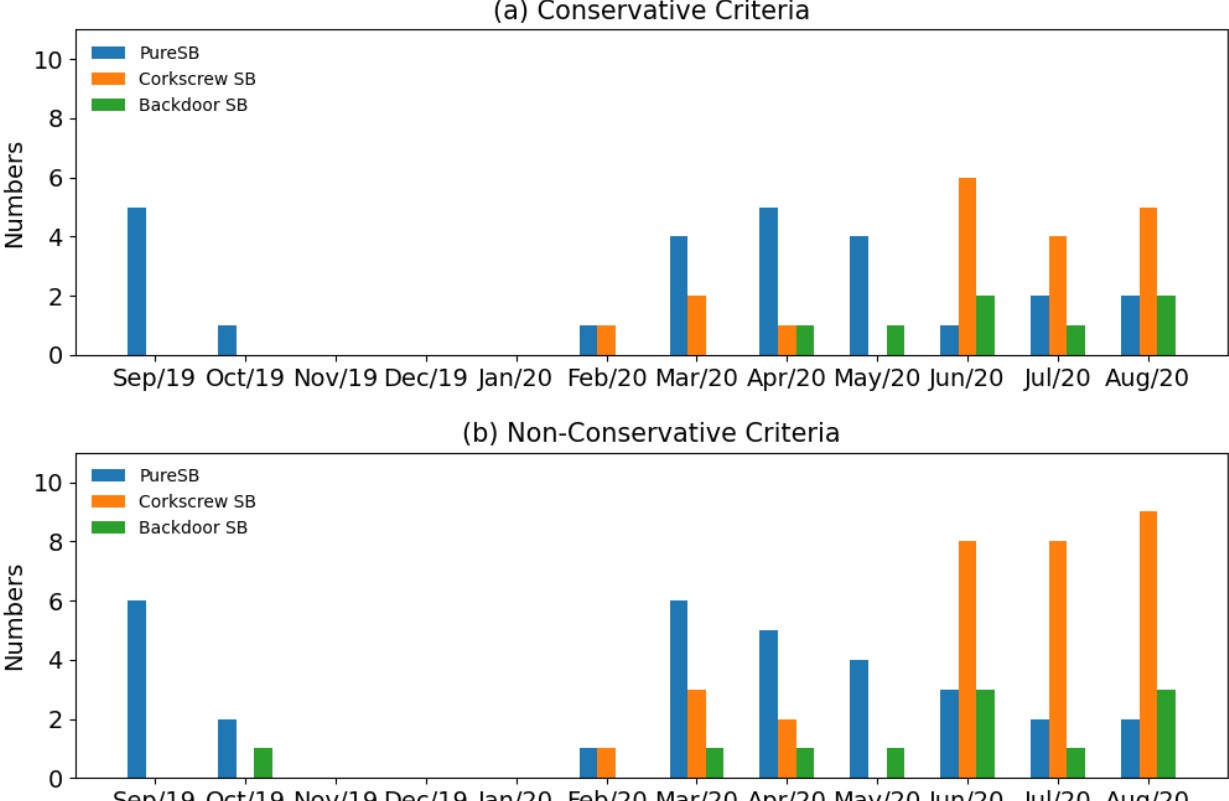

**Figure 10:** Monthly timeseries of identified sea breeze events from September 2019 to August 2020 using the conservative criteria (a) and non-conservative criteria (b) for the pure sea breezes (blue), the corkscrew sea breezes (orange) and the backdoor sea breezes (green).

The offshore wind energy potential associated with each type of sea breeze is first assessed by examining the spatial pattern of hub-height (100 meters above ground level) level wind in the sea breeze composites (Figure 11). At the hub-height level, the general features of the different types of sea breezes, such as the calm zone, coastal jet, veering (backing) of the wind, are still present. Furthermore, similar wind pattens are also apparent at the 200-m level (figure not shown). This suggests that the rotor-layer area of the offshore wind turbine are subject to potential impact from the sea breeze events. To estimate the potential impact of sea breezes on offshore wind power, power output is calculated using the power coefficient associated with a IEA 10 Megawatt (MW) reference offshore wind turbine is used. Figure 12a) shows the timeseries of hub-height wind speed associated with each type of sea breeze near the coast (at center point of the targeted region in Figure 1b). During a pure sea breeze event, the hub-height wind speed decreases during the morning hours due to the development of the calm zone and recovers back to approximately 7-8 m/s in the afternoon. In the case of the backdoor sea breeze, the strongest hub-height wind speed (8-9 m/s) occurs in the morning and then starts to decrease through the late afternoon. In contrast, the hub-

height wind speed associated with the corkscrew sea breeze is the weakest in the morning, after which it increases as the sea

breeze develops, and finally reaches its maximum value (10-11 m/s) around the late afternoon. Figure 12b) shows the associated power output for each type of sea breeze. The most significant difference among the cases occurs in the afternoon, when the coastal jet associated with the corkscrew sea breezes forms, and the associated power output is almost 3 to 4 times as large as that of the pure and backdoor sea breeze cases. Note that the backdoor sea breeze is associated with the highest power output during the morning hours. Selecting a different reference point, either closer or further away from the coast,

does not change the overall pattern of the results. This highlights the importance of the predicting the correct type of sea breeze, especially the corkscrew sea breeze, for the wind energy application. In addition, the layout and positioning of the wind farm might also have a significant impact on the power output during a sea breeze event. For instance, a wind farm might be split by the calm zone but has more power production due to less wake loss. Therefore, finding the right layout of wind farm is also important for offshore wind energy.

Other factors such as changes in the WRF configuration, statistical approach and targeted region, could have potential sensitivity to the overall number as well as the seasonal distribution for each type of sea breeze. Nevertheless, the importance of identifying the correct type of sea breeze for wind energy forecast would still be significant and serve as a high-priority research topic, especially for offshore wind energy.


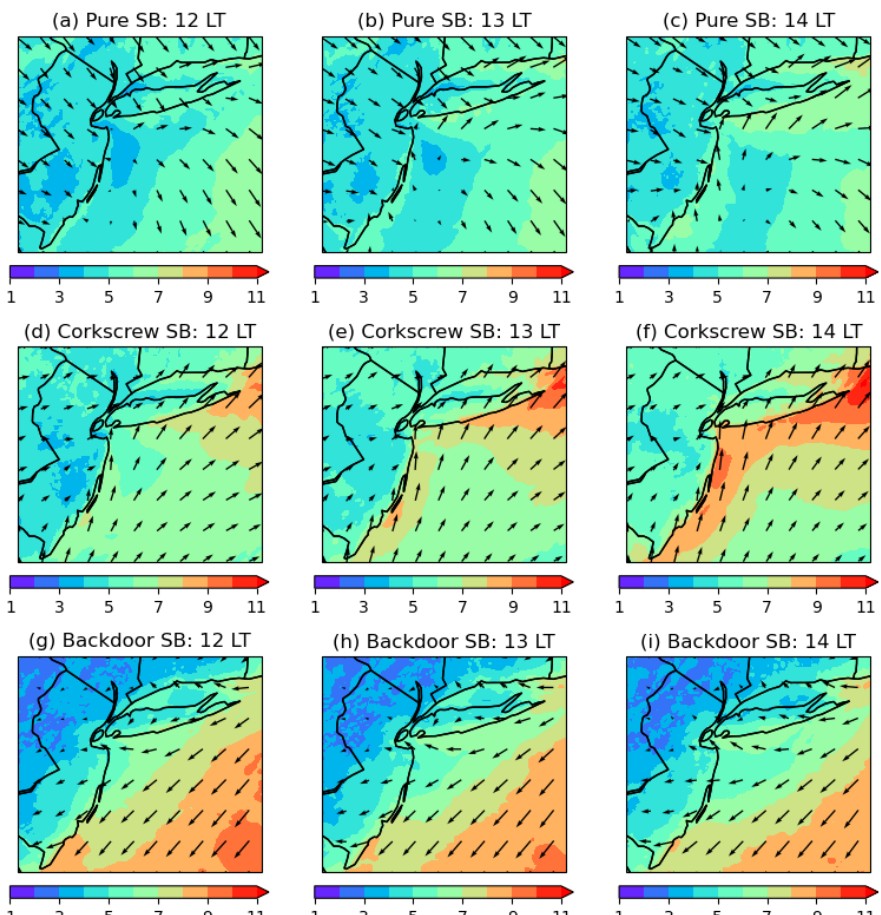

**Figure 11**: Spatial pattern of hub-height level wind speed and wind direction from the composite average for the pure sea breezes (a-c), corkscrew sea breezes (d-f) and backdoor sea breezes (g-i) at 12, 13 and 14 local time (LT). The colored contours indicate the magnitude of wind speed (m/s) and the vectors show the wind direction.

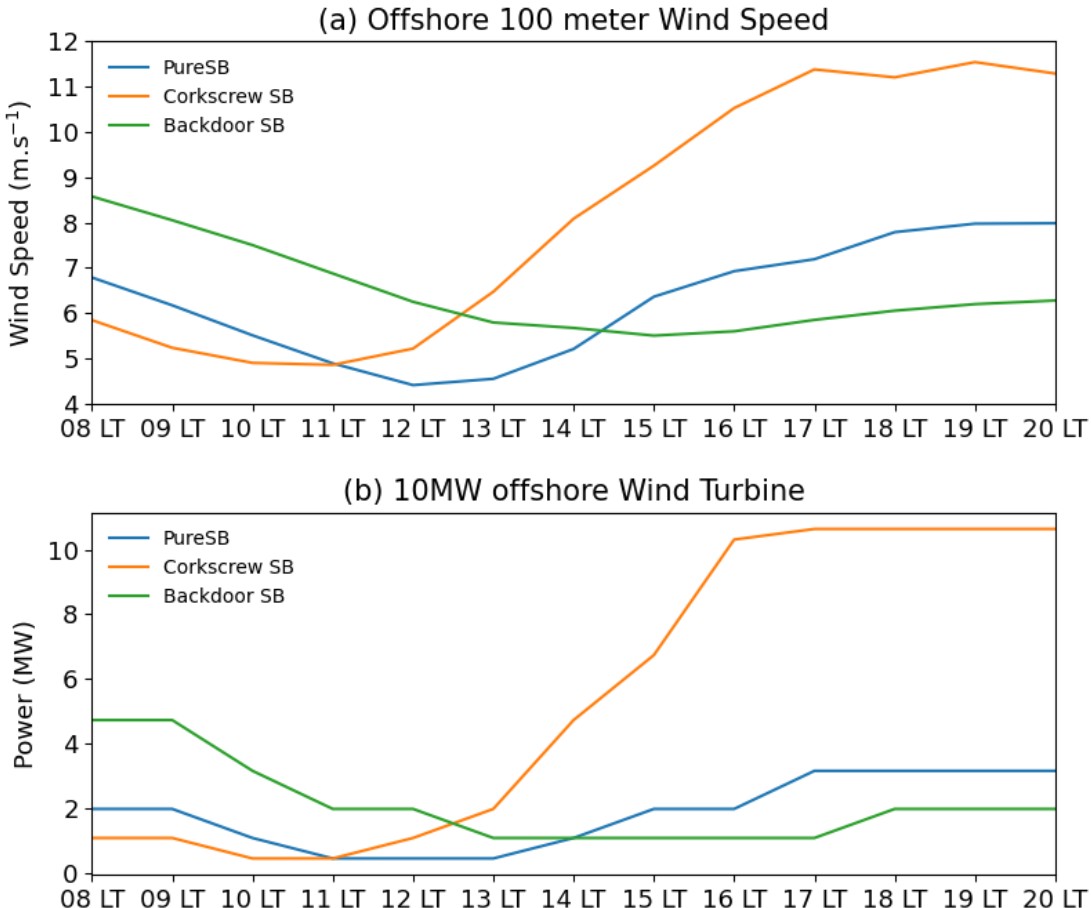

**Figure 12:** Timeseries of (a) offshore hub-height wind speed (m/s) and (b) the associated power output from a 10 MW wind turbine with respect to sea breeze type between 08 LT and 20 LT.


Even though the focus of this paper is not to validate the simulated sea breezes with observations, some characteristics associated with the modeled sea breezes do differ from observations documented in the past literatures. For instance, due to the nature of forced balance between local sea breeze circulations and synoptic scale forcing, the timing of landfall arrival for each type of sea breeze should vary among cases, with the corkscrew sea breeze being the earliest and the backdoor sea

breeze being the latest (Adam, 1997, Miller et al. 2003). However, such feature is not evident in this study. The inland convergence line forms around noon time for all three cases. In addition, the sea breeze cycle starts, develops and manifests around the same time regardless of the type (Figures 6-8). This suggests some potential limitations with the current numerical model (WRF) in simulating sea breezes. For example, the current model setup lacks ocean-atmosphere dynamic coupling. Correctly modeling sea breeze development strongly depends on accurate ocean-atmosphere heat fluxes and

momentum exchanges, so the absence of such coupling can introduce errors. Therefore, developing a dynamical ocean

model that couples with WRF would be, in principle, beneficial to simulating sea breezes and providing more valuable environmental assessments for offshore wind energy.

**5. Conclusion**

With the vastly planned offshore wind farm construction along the U.S. East coast, identifying and understanding key coastal process, such as sea breezes, has become a critical need for the sustainability and development of the U.S. offshore wind energy. In this study, a new two-step identification method has been proposed to detect and characterize three types of sea breezes (pure, corkscrew and backdoor) from a year-long WRF simulation.

The results suggest that the proposed detection method is able to identify different types of sea breeze events in the model simulations. Key sea breeze features, such as the calm zone associated with pure sea breezes and the coastal jet associated with corkscrew sea breezes, are evident in the sea breeze composite output. In addition, the simulated sea breeze events indicate a transition from pure sea breezes to corkscrew sea breezes during the warm season as the land-sea thermal contrast increases. Furthermore, the location and extension of sea breeze fronts is different for each type of sea breeze, suggesting

that the coastal impact of sea breeze varies with sea breeze type. Using the power coefficient from a 10 MW offshore wind turbine, the power production associated with the corkscrew sea breeze is almost 3 to 4 times larger than the power generated by other two types of sea breezes. This points out the importance of forecasting the correct type of sea breeze, especially the corkscrew sea breeze, in numerical weather/wind energy forecasting.

Even though this is not a validation study, the modeled sea breeze events do show discrepancy with the observations from the past literatures in certain aspects, such as the landfall timing. There are many possible reasons for this; one potential source is the lack of ocean-atmosphere dynamic coupling in the current model setup. This motivates the development and usage of a fully coupled ocean-atmosphere model in sea breeze research, which will be the focus of a subsequent study.

**Code/Data availability**

OSTIA data are publicly available from the Physical Oceanography Distributed Active Archive Center. ERA-5 data are publicly available from the ECMWF's MARS archive. The WRF simulations and code are available from the lead author upon request.

**Author contributions**

Geng Xia – prepared the manuscript with contributions from all co-authors; conceptualization, formal analysis, investigation, validation, visualization, writing (original draft, review and editing)

Caroline Draxl– conceptualization, project administration, supervision, writing (review and editing)

Michael Optis – conceptualization, designing and conducting simulation

Stephanie Redfern – conceptualization, writing (review and editing)


**Competing interests**

The authors declare that they have no conflict of interest.

**Acknowledgements**

The research was performed using computational resources sponsored by the Department of Energy's Office of Energy Efficiency and Renewable Energy and located at the National Renewable Energy Laboratory. This work was authored in part by the National Renewable Energy Laboratory, operated by Alliance for Sustainable Energy, LLC, for the U.S. Department of Energy (DOE) under Contract No. DE-AC36-08GO28308. Funding was provided by the U.S. Department of Energy Office of Energy Efficiency and Renewable Energy Wind Energy Technologies Office. The views expressed in the article do 425 not necessarily represent the views of the DOE or the U.S. Government. The U.S. Government retains and the publisher, by accepting the article for publication, acknowledges that the U.S. Government retains a nonexclusive, paid-up, irrevocable, worldwide license to publish or reproduce the published form of this work, or allow others to do so, for U.S. Government purposes. A portion of the research was performed using computational resources sponsored by the Department of Energy's Office of Energy Efficiency and Renewable Energy and located at the National Renewable Energy Laboratory.

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
