# Peer review of "Detecting and Characterizing Simulated Sea Breezes Over the U.S. Northeast Coast with Implication for Offshore Wind Energy"

_Wind Energy Science, 2021_

## Author Comment (AC1)

**General Comments**
With the progress in the floating turbines technology, an increasing number of offshore projects are going to be implemented in coastal deep waters; offshore areas so far disregarded, are receiving now increasing attention like the US East coast.

In this context, this paper is timely focusing on the major meteorological phenomenon in coastal areas i,e, the breezes. It presents a methodology to detect the three types of sea breezes and their characteristic features, such as the calm zone associated with pure sea breezes and coastal jets associated with corkscrew sea breezes and discuss those feature in a wind energy prospective

Response: Thank you for your positive views on our paper. We sincerely appreciate the time you spent reviewing this work. In this revision, we have revised the paper substantially based on yours and other reviewers' comments. The key changes are
- The title of the paper has changed to *Detecting and Characterizing **Simulated** Sea Breezes Over the U.S. Northeast Coast with Implication for Offshore Wind Energy*.
- An additional analysis has been conducted to examine the variability of individual sea breeze cases.

Specific Comments
This paper is well structured and written but, in my opinion, the authors, should expand the discussion of the impact of the SB from the wind energy perspective. In fact, the authors show that there are calms and divergence zone that impact on single turbine production in different breeze types (pure and corkscrew and backdoor).

They found that "the power production associated with a 10 megawatts offshore wind turbine would produce approximately 3 to 4 times more electrical power during a corkscrew sea breeze event than the other two types of sea breezes". But there is more than this. There is the issue of finding the right layout of a wind farm or of wind farms clusters with respect to wakes; a wind farm might be split by a calm zone in at least two areas with different wind directions. In this case, the wake losses of the whole wind farm might be less and the production more.

Response: Thank you for your comment. You are right. The layout of the wind farm clusters will have a significant impact on the overall power output. However, to analyze such impact, WRF simulation with wind farm parameterization will be needed. That is beyond the scope of this study but will be serve as an interesting topic for the future work. Nevertheless, I have added a few sentences in the revised manuscript to discuss this matter.

*"In addition, the layout and positioning of the wind farm might have a significant impact on the power output during a sea breeze event. For instance, a wind farm might be split by the calm zone but has more power production due to less wake loss. Therefore, finding the right layout of wind farm is also important for offshore wind energy."*

@pag 10 the authors write " In addition, the location of the calm zone varies by cases, although most calm zones develop relatively close to the coastline " Here, my comment is that an analysis of the variability of the distance from the cost and the amplitude of the calm zone are variables s for sure of interest for projects developers.

Response: Thank you for your comment. We have conducted additional analysis to examine the variability of simulated sea breeze events to address your concern. Our results suggest that the temporal development of the calm zone for the pure sea breeze and the positioning of the coastal jet for the corkscrew sea breeze is rather consistent across their identified cases respectively.

[Figure]

To do that, we have defined three regions to quantify the variability of the identified sea breeze cases (as shown in the figure). They are located on land (blue), over the coast (red) and over the ocean (green). The size of region is about 3 % of the entire regional domain. For each sea breeze type, we calculate the standard deviation of WS10 and WD10 from the identified sea breeze events over all three regions from 08 LT to 20 LT, and the results are shown in the tables below.

Table1 : Variability of simulated pure sea breeze cases over land, coast region and ocean

| | 08 LT | 09 LT | 10 LT | 11 LT | 12 LT | 13 LT | 14 LT | 15 LT | 16 LT | 17 LT | 18 LT | 19 LT | 20 LT |
|---|---|---|---|---|---|---|---|---|---|---|---|---|---|
| Standard Deviation of WS10(m/s) for the Identified Pure Sea Breeze Cases | | | | | | | | | | | | | |
| Inland | 1.4 | 1.3 | 1.2 | 1.1 | 1.2 | 1.3 | 1.4 | 1.3 | 1.1 | 0.9 | 1.0 | 0.8 | 0.8 |
| Coast | 2.3 | 2.0 | 1.8 | 1.6 | 1.5 | 1.4 | 1.3 | 1.4 | 1.4 | 1.4 | 1.3 | 1.2 | 1.4 |
| Ocean | 3.0 | 2.8 | 2.5 | 2.1 | 1.8 | 1.7 | 1.7 | 1.6 | 1.5 | 1.6 | 1.7 | 1.8 | 1.8 |
| Standard Deviation of WD10 (degree) for the Identified Pure Sea Breeze Cases | | | | | | | | | | | | | |
| Inland | 119 | 111 | 94 | 77 | 62 | 58 | 58 | 62 | 63 | 63 | 56 | 51 | 55 |
| Coast | 102 | 119 | 114 | 105 | 92 | 78 | 65 | 60 | 55 | 54 | 53 | 62 | 51 |
| Ocean | 118 | 129 | 112 | 108 | 107 | 116 | 115 | 110 | 106 | 97 | 85 | 78 | 76 |

For the pure sea breeze cases (Table1), the variability of WS10 is largest during the morning hours and decreases after that. Overall, the variable of WS10 is greater over the ocean than that on land. As for WD10, the variability is large during the morning hours. Note that, based on our methodology and the shape of the coastline, the pure sea breeze is identified from potential days of three different wind regimes (Northwesterly, Northly and Westly). Therefore, it is not a surprise that variability of WD10 is large during the morning hour. However, variability of WD10 drastically decreases after the morning hour due to the influence of sea breeze development. Note that the standard deviation of WD10 over the ocean is relatively large until late afternoon. This is mainly due to the development of the calm zone (Figure 6 of the manuscript). After the calm zone moved away from the coast, standard deviation of WD10 reduces significantly (16 LT to 20 LT).

Table2 : Variability of simulated corkscrew sea breeze cases over land, coast region and ocean

Standard Deviation of WS10(m/s) for the Identified Pure Sea Breeze Cases

| | 08 LT | 09 LT | 10 LT | 11 LT | 12 LT | 13 LT | 14 LT | 15 LT | 16 LT | 17 LT | 18 LT | 19 LT | 20 LT |
|---|---|---|---|---|---|---|---|---|---|---|---|---|---|
| Inland | 1.1 | 1.0 | 1.1 | 1.1 | 1.1 | 1.0 | 0.9 | 0.8 | 0.6 | 0.8 | 1.0 | 0.6 | 0.7 |
| Coast | 1.8 | 1.7 | 1.6 | 1.7 | 1.9 | 1.7 | 1.8 | 1.7 | 1.8 | 1.8 | 1.7 | 1.6 | 1.8 |
| Ocean | 2.4 | 2.3 | 2.1 | 1.9 | 1.8 | 1.9 | 2.1 | 2.2 | 2.3 | 2.5 | 2.4 | 2.4 | 2.2 |

Standard Deviation of WD10 (degree) for the Identified Pure Sea Breeze Cases

| | 08 LT | 09 LT | 10 LT | 11 LT | 12 LT | 13 LT | 14 LT | 15 LT | 16 LT | 17 LT | 18 LT | 19 LT | 20 LT |
|---|---|---|---|---|---|---|---|---|---|---|---|---|---|
| Inland | 68 | 67 | 58 | 48 | 42 | 43 | 44 | 43 | 43 | 44 | 40 | 46 | 40 |
| Coast | 79 | 80 | 69 | 55 | 45 | 36 | 32 | 35 | 28 | 28 | 28 | 35 | 31 |
| Ocean | 67 | 78 | 85 | 75 | 75 | 78 | 66 | 62 | 60 | 57 | 48 | 36 | 31 |

Table 2 shows the results from the corkscrew sea breezes. In general, the characteristics are similar to that from the pure sea breeze cases. One important aspect is that the small variability of WD10 over the coastal region during the late afternoon hours. This suggests that the position of the simulated jet core (Figure 7 of the manuscript) over this region is rather stable, which would have significant offshore wind energy implication in terms of wind turbine positioning.

Table3 : Variability of simulated backdoor sea breeze cases over land, coast region and ocean

Standard Deviation of WS10(m/s) for the Identified Pure Sea Breeze Cases

| | 08 LT | 09 LT | 10 LT | 11 LT | 12 LT | 13 LT | 14 LT | 15 LT | 16 LT | 17 LT | 18 LT | 19 LT | 20 LT |
|---|---|---|---|---|---|---|---|---|---|---|---|---|---|
| Inland | 0.9 | 1.1 | 1.6 | 1.7 | 1.6 | 1.7 | 1.7 | 1.7 | 1.8 | 1.4 | 0.8 | 0.4 | 0.3 |
| Coast | 2.4 | 2.3 | 2.2 | 2.4 | 2.5 | 2.4 | 2.3 | 2.3 | 2.3 | 2.3 | 2.2 | 2.3 | 2.4 |
| Ocean | 3.5 | 3.7 | 3.6 | 3.5 | 3.4 | 3.0 | 2.9 | 3.0 | 3.0 | 3.0 | 2.9 | 2.8 | 2.7 |

Standard Deviation of WD10 (degree) for the Identified Pure Sea Breeze Cases

| | 08 LT | 09 LT | 10 LT | 11 LT | 12 LT | 13 LT | 14 LT | 15 LT | 16 LT | 17 LT | 18 LT | 19 LT | 20 LT |
|---|---|---|---|---|---|---|---|---|---|---|---|---|---|
| Inland | 16 | 17 | 20 | 29 | 34 | 37 | 35 | 31 | 31 | 27 | 27 | 29 | 31 |
| Coast | 13 | 14 | 17 | 18 | 16 | 16 | 17 | 16 | 18 | 22 | 22 | 28 | 26 |
| Ocean | 107 | 94 | 42 | 37 | 58 | 65 | 78 | 62 | 36 | 24 | 17 | 22 | 29 |

Table 3 shows the results from the backdoor sea breezes. Because of low occurrence rate, It has the smallest variability, which also indicates that the development of the individual backdoor sea breeze does not differ much from the mean condition (Figure 8 of the manuscript).

Corresponding texts and tables have been added to the manuscript. Note that we did change the alignment of three regions in other attempts, such as horizontal and vertical. However, that does not have a significant impact on the results.

---

## Author Comment (AC2)

The paper is well written and easy to read. It considers an important topic for wind energy meteorology and better understanding of sea-breezes is a welcome addition to the field. However, I have a few comments considering the methodology. I believe that there are some critical aspects of methodology that have not been properly described.

Response: Thank you for your positive views on our paper. We sincerely appreciate the time you spent reviewing this work. In this revision, we have revised the paper substantially based on yours and other reviewer's comment. The key changes are

- The title of the paper has changed to *Detecting and Characterizing **Simulated** Sea Breezes Over the U.S. Northeast Coast with Implication for Offshore Wind Energy*.
- An additional analysis has been conducted to examine the variability of individual sea breeze cases.

Major comments.

- P3L84-L85. The simulations are one month long. Was there any kind of nudging performed during the simulations? If not, please explain why there was no nudging performed because one month is quite a long time and the model can "run away" from the real atmospheric conditions.

Response: Thank you for your comment. Yes, atmospheric nudging is applied on the outer domain every 6 hours. Corresponding information has been added to the revised manuscript.

- P6L129. Mean wind direction at 10 m is calculated. Wind direction is a circular variable and therefore I find it hard to interpret "mean wind direction". Please explain in more detail how you calculated mean wind direction in complicated meteorological situations and why such an approach is feasible, especially, taking into account the fact that the averaging is done over coastal quadrants, where sea breeze front can be present and therefore opposite wind directions can be next to each other. For instance, the average of W and E direction is S wind.

Response: Thank you for your comment. The mean wind direction is calculated for all the four quadrants by simply averaging all the points within the quadrant. Then, the wind regime over the targeted region is determined if the mean wind direction for at least 2 of the 4 quadrants fall under one of the five wind regime categories.

We understand your concern *('the average of W and E direction is S wind')* which would be an issue if we did the averaging over just a few points. However, what we did is a regional average which will not be affected by a few outliers. For a spatial extent of a quadrant size domain defined in our study, very rarely, will you see wind direction changes drastically from 0° to 180° from one quadrant to the other. If that happens, that usually mean the region is influenced by cyclonic conditions and those cases are filtered out in the first place (Lines 129-130: *Days with cyclonic conditions over the targeted region are rejected, as sea breeze*

*identification would be difficult due to rapid changes in wind direction in these cases*). In addition, I have checked all the selected days as well as all the unselected days to make sure our method works.

Figure 3. I do not understand Figure 3. It is supposed to demonstrate the differences in prevailing wind between different types of sea-breeze. But the classification is based on the relationship between the prevailing wind and the shoreline. In these schematics the shoreline is not indicated. The brown and blue color is especially confusing here, because it is reminiscent of land/sea border in maps. I also have a problem that authors haven't defined how they interpret the direction of shoreline in the actual map. Is the north wind supposed to be purely offshore (coast-perpendicular) wind in this study? But the coastline is not oriented W-E, it has a complicated shape. Authors should clearly describe how they interpret the shoreline direction in this study and which prevailing wind directions correspond to "pure", "corkscrew" and "backdoor" directions and why.

Response: Thank you for your comment and sorry for the confusion. First of all, we did not re-define shoreline. The shoreline is the coastline of our targeted region (See figure below). As you also stated, the shoreline in our case has a complicated shape. Therefore, the three schematics show the *idealized* sea level pressure conditions and their associated prevailing conditions based on the shoreline of the targeted region. Because of the complicated shape of the shoreline, we have to separate the wind regimes into five categories to better facilitate the identification of sea breeze type in the second step. For each day, the SLP condition from the four quadrants is first used to determine the potential background prevailing wind (see figure) before the rigorous classification of wind regime takes place.

[Figure]

- Figure 6. I am not sure if showing the composites here is the best way how to represent the findings. Authors admit it themselves: "Even though the composite 10 wind speed over the calm zone is between 2 and 4 m/s, it falls primarily between 0 and 1 m/s for each individual case". I am wondering if showing a representative single case would not be better to illustrate the properties of sea breeze. I am wondering whether the problem is the fact that the evolution of sea-breeze depends less on the "absolute" timing and more on the hours elapsed after sunrise. Maybe if the composite was done by averaging timeframes relative to the time after sunrise, the composites would be better. I imagine that sunrise time changes quite a lot during the year at those latitudes.

Response: Thank you for your comment. I agree with your concern about using the composites to represent the finding (Other reviewers have mentioned this as well). However, I also don't think using a single case is a good idea because it is difficult to objectively choose which case to present in the paper and I am sure there will be people questioning that as well. I think the best way is to describe the variability of the identified sea breeze events from the model simulations.

In this revision, we have conducted additional analysis to examine the variability of simulated sea breeze events to address your concern. Our results suggest that the temporal development of the calm zone for the pure sea breeze and the positioning of the coastal jet for the corkscrew sea breeze is rather consistent across their identified cases respectively.

[Figure]

To do that, we have defined three regions to quantify the variability of the identified sea breeze cases (as shown in the figure). They are located on land (blue), over the coast (red) and over the ocean (green). The size of region is about 3 % of the entire regional domain. For each sea breeze type, we calculate the standard deviation of WS10 and WD10 from the identified sea breeze events over all three regions from 08 LT to 20 LT, and the results are shown in the tables below.

Table1 : Variability of simulated pure sea breeze cases over land, coast region and ocean

| Standard Deviation of WS10(m/s) for the Identified Pure Sea Breeze Cases | | | | | | | | | | | | |
|---|---|---|---|---|---|---|---|---|---|---|---|---|
| 08 LT | 09 LT | 10 LT | 11 LT | 12 LT | 13 LT | 14 LT | 15 LT | 16 LT | 17 LT | 18 LT | 19 LT | 20 LT |

| | 08 LT | 09 LT | 10 LT | 11 LT | 12 LT | 13 LT | 14 LT | 15 LT | 16 LT | 17 LT | 18 LT | 19 LT | 20 LT |
|---|---|---|---|---|---|---|---|---|---|---|---|---|---|
| Inland | 1.4 | 1.3 | 1.2 | 1.1 | 1.2 | 1.3 | 1.4 | 1.3 | 1.1 | 0.9 | 1.0 | 0.8 | 0.8 |
| Coast | 2.3 | 2.0 | 1.8 | 1.6 | 1.5 | 1.4 | 1.3 | 1.4 | 1.4 | 1.4 | 1.3 | 1.2 | 1.4 |
| Ocean | 3.0 | 2.8 | 2.5 | 2.1 | 1.8 | 1.7 | 1.7 | 1.6 | 1.5 | 1.6 | 1.7 | 1.8 | 1.8 |

| Standard Deviation of WD10 (degree) for the Identified Pure Sea Breeze Cases | | | | | | | | | | | | | |
|---|---|---|---|---|---|---|---|---|---|---|---|---|---|
| | 08 LT | 09 LT | 10 LT | 11 LT | 12 LT | 13 LT | 14 LT | 15 LT | 16 LT | 17 LT | 18 LT | 19 LT | 20 LT |
| Inland | 119 | 111 | 94 | 77 | 62 | 58 | 58 | 62 | 63 | 63 | 56 | 51 | 55 |
| Coast | 102 | 119 | 114 | 105 | 92 | 78 | 65 | 60 | 55 | 54 | 53 | 62 | 51 |
| Ocean | 118 | 129 | 112 | 108 | 107 | 116 | 115 | 110 | 106 | 97 | 85 | 78 | 76 |

For the pure sea breeze cases (Table1), the variability of WS10 is largest during the morning hours and decreases after that. Overall, the variable of WS10 is greater over the ocean than that on land. As for WD10, the variability is large during the morning hours. Note that, based on our methodology and the shape of the coastline, the pure sea breeze is identified from potential days of three different wind regimes (Northwesterly, Northly and Westly). Therefore, it is not a surprise that variability of WD10 is large during the morning hour. However, variability of WD10 drastically decreases after the morning hour due to the influence of sea breeze development. Note that the standard deviation of WD10 over the ocean is relatively large until late afternoon. This is mainly due to the development of the calm zone (Figure 6 of the manuscript). After the calm zone moved away from the coast, standard deviation of WD10 reduces significantly (16 LT to 20 LT).

Table2 : Variability of simulated corkscrew sea breeze cases over land, coast region and ocean

| Standard Deviation of WS10(m/s) for the Identified Pure Sea Breeze Cases | | | | | | | | | | | | | |
|---|---|---|---|---|---|---|---|---|---|---|---|---|---|
| | 08 LT | 09 LT | 10 LT | 11 LT | 12 LT | 13 LT | 14 LT | 15 LT | 16 LT | 17 LT | 18 LT | 19 LT | 20 LT |
| Inland | 1.1 | 1.0 | 1.1 | 1.1 | 1.1 | 1.0 | 0.9 | 0.8 | 0.6 | 0.8 | 1.0 | 0.6 | 0.7 |
| Coast | 1.8 | 1.7 | 1.6 | 1.7 | 1.9 | 1.7 | 1.8 | 1.7 | 1.8 | 1.8 | 1.7 | 1.6 | 1.8 |
| Ocean | 2.4 | 2.3 | 2.1 | 1.9 | 1.8 | 1.9 | 2.1 | 2.2 | 2.3 | 2.5 | 2.4 | 2.4 | 2.2 |

| Standard Deviation of WD10 (degree) for the Identified Pure Sea Breeze Cases | | | | | | | | | | | | | |
|---|---|---|---|---|---|---|---|---|---|---|---|---|---|
| | 08 LT | 09 LT | 10 LT | 11 LT | 12 LT | 13 LT | 14 LT | 15 LT | 16 LT | 17 LT | 18 LT | 19 LT | 20 LT |
| Inland | 68 | 67 | 58 | 48 | 42 | 43 | 44 | 43 | 43 | 44 | 40 | 46 | 40 |
| Coast | 79 | 80 | 69 | 55 | 45 | 36 | 32 | 35 | 28 | 28 | 28 | 35 | 31 |
| Ocean | 67 | 78 | 85 | 75 | 75 | 78 | 66 | 62 | 60 | 57 | 48 | 36 | 31 |

Table 2 shows the results from the corkscrew sea breezes. In general, the characteristics are similar to that from the pure sea breeze cases. One important aspect is that the small variability of WD10 over the coast region during the late afternoon hours. This suggests that the position of the simulated jet core (Figure 7 of the manuscript) over this region is rather stable, which would have significant offshore wind energy implication in terms of wind turbine positioning.

Table3 : Variability of simulated backdoor sea breeze cases over land, coast region and ocean

| Standard Deviation of WS10(m/s) for the Identified Pure Sea Breeze Cases | | | | | | | | | | | | | |
|---|---|---|---|---|---|---|---|---|---|---|---|---|---|
| | 08 LT | 09 LT | 10 LT | 11 LT | 12 LT | 13 LT | 14 LT | 15 LT | 16 LT | 17 LT | 18 LT | 19 LT | 20 LT |
| Inland | 0.9 | 1.1 | 1.6 | 1.7 | 1.6 | 1.7 | 1.7 | 1.7 | 1.8 | 1.4 | 0.8 | 0.4 | 0.3 |

| | 08 LT | 09 LT | 10 LT | 11 LT | 12 LT | 13 LT | 14 LT | 15 LT | 16 LT | 17 LT | 18 LT | 19 LT | 20 LT |
|---|---|---|---|---|---|---|---|---|---|---|---|---|---|
| Coast | 2.4 | 2.3 | 2.2 | 2.4 | 2.5 | 2.4 | 2.3 | 2.3 | 2.3 | 2.3 | 2.2 | 2.3 | 2.4 |
| Ocean | 3.5 | 3.7 | 3.6 | 3.5 | 3.4 | 3.0 | 2.9 | 3.0 | 3.0 | 3.0 | 2.9 | 2.8 | 2.7 |

| Standard Deviation of WD10 (degree) for the Identified Pure Sea Breeze Cases | | | | | | | | | | | | | |
|---|---|---|---|---|---|---|---|---|---|---|---|---|---|
| | 08 LT | 09 LT | 10 LT | 11 LT | 12 LT | 13 LT | 14 LT | 15 LT | 16 LT | 17 LT | 18 LT | 19 LT | 20 LT |
| Inland | 16 | 17 | 20 | 29 | 34 | 37 | 35 | 31 | 31 | 27 | 27 | 29 | 31 |
| Coast | 13 | 14 | 17 | 18 | 16 | 16 | 17 | 16 | 18 | 22 | 22 | 28 | 26 |
| Ocean | 107 | 94 | 42 | 37 | 58 | 65 | 78 | 62 | 36 | 24 | 17 | 22 | 29 |

Table 3 shows the results from the backdoor sea breezes. Because of low occurrence rate, It has the smallest variability, which also indicates that the development of the individual backdoor sea breeze does not differ much from the mean condition (Figure 8 of the manuscript).

Corresponding texts and tables have been added to the manuscript. Note that we have changed the alignment of three regions in other attempts, such as horizontal and vertical. However, that does not change the table results significantly.

- P9L200-202 " This could be partially associated with the increase in the land-sea thermal contrast. As the land-sea temperature difference becomes more strongly positive, there is greater potential for corkscrew sea breeze development over pure sea breeze development along the U.S. Northeast coast." I am confused about such an assertion. If the difference between pure and corkscrew sea breezes comes from the difference in prevailing wind direction (some authors use "geostrophic wind" here), how does the sea-land temperature difference influence prevailing wind direction?

Response: Thank you for your comment. I certainly agree with you that the prevailing wind direction is the dominant factor in differentiating the pure and corkscrew sea breeze. In addition, the prevailing wind direction is mostly determined by the large-scale forcing, rather than the sea-land temperature difference. However, the sea-land temperature difference does provide the forcing for sea breeze formation. In general, the greater the temperature difference, the stronger the sea breeze. This will lead to stronger onshore wind. As the strong onshore wind is the distinct feature associated with the corkscrew sea breeze, we hypothesize that the stronger sea-land temperature difference might help generate the strong onshore wind which increases the likelihood of forming corkscrew sea breeze, rather than pure sea breeze, during the summer months. Nevertheless, this hypothesis needs to be further proven and is beyond the scope of this study. That is why we use "*partially associated*" in our sentence to acknowledge the uncertainty.

Minor comments:

- Figure 2. It would be easier to understand Figure 2 if the explanations for the abbreviations, such as WR: "Wind regime" would be explained in the figure caption.

Response: Thank you for your comment. Corresponding changes have been made.

- Line 152. "SW-WR" – I assume that "CS-WR" is meant here.

Response: Thank you for your comment. Corresponding changes have been made.

---

## Author Comment (AC3)

The paper described a method to identify three types of sea breezes – pure, backdoor, and corkscrew – from WRF model simulations conducted over the US Northeast. The simulations cover only one year (September 2019 – August 2020) and used two-way nested domains of 6 and 2 km resolution. Statistics of the results are presented, as well as a sensitivity analysis to the values of the thresholds adopted to identify the sea breezes and a comparison of the typical wind power production to be expected for each sea breeze type.

The paper is well written and the figures are clear, but there are so many issues with the paper that, in my opinion, it should be rejected for publication in this journal, although it might be suitable somewhere else.

Response: Thank you for your comment. We sincerely appreciate the time you spent reviewing this work. In this revision, we have revised the paper substantially based on yours and other reviewer's comment. The key changes are
- The title of the paper has changed to *Detecting and Characterizing **Simulated** Sea Breezes Over the U.S. Northeast Coast with Implication for Offshore Wind Energy*.
- An additional analysis has been conducted to examine the variability of individual sea breeze cases.

I'll focus on major issues first.

The first major issue is that it is not clear why this paper is relevant for wind energy. The authors focus on sea breezes along the US East Coast and only at the end of the Discussion section present two figures somewhat relevant to offshore wind to presumably show that the power output depends on the sea breeze type. How innovative or useful is this type of information? Why was the turbine placed in the center of domain 2? How would a developer or wind farm operator benefit from this information? I suggest that the authors choose a different journal, one perhaps focused on climatological or meteorological aspects.

Response: Thank you for your comment. This paper is not a validation study, it is to introduce a method to detect sea breeze events over the U.S Northeast Coast from WRF simulation. In this revision, we have changed the title to be better reflect our purpose. To our best knowledge, this is the second paper to develop numerical method for sea breeze detection from WRF simulation and it is the first to apply over the U.S. Northeast Coast where there is huge offshore wind potential (https://www.northeastoceandata.org/data-explorer/?energy-infrastructure|planning-areas). As sea breezes are a significant coastal phenomenon, detecting sea breezes from numerical model will be relevant to the wind energy community and offshore wind forecasting.

The second issue is that the simulations cover only one year, therefore they are not long enough to produce meaningful statistics, climatologically speaking. The paper does not explain why the period was chosen. I would understand if the authors had collected observations over that period and wanted to validate the model results, but

they did not, which is in fact my third issue. There is no model validation and we are left with no convincing evidence that the two-step method indeed finds sea breeze events correctly.

Response: Thank you for your comment. As we mentioned in the Major comment 1, model validation is not the focus of this study; rather it is to develop a method to detect sea breeze. Therefore, we believe a year-long simulation is enough to demonstrate our method so that others can use the method and apply in their long-term simulation. This work started off in 2019 and that is why we run the simulation from 2019 to 2020. The validity of our method is shown in the composite figures which demonstrated the key structures and features associated with each type of sea breeze.

The fourth issue is methodological. The paper uses averages and means abundantly. I am particularly troubled by the use of average wind directions. Since the wind is a vector, the sum of vectors is not an average. What's the average of a northerly (0°) and southerly (180°) wind? A wind from the east (90°)? It does not make any physical sense. As such, Figures 4, 6-8, and 11 are not acceptable because they show "average" wind vectors.

Response: Thank you for your comment. I understand your concern which would be an issue if we did the averaging over just a few points. However, what we did is a regional average which will not be affected by a few outliers. For a spatial extent of a quadrant size domain defined in our study, very rarely, will you see wind direction changes drastically from 0° to 180 ° from one quadrant to the other. If that happens, that usually mean the region is influenced by cyclonic conditions and those cases are filtered out in the first place (Lines 129-130: *Days with cyclonic conditions over the targeted region are rejected, as sea breeze identification would be difficult due to rapid changes in wind direction in these cases*). In addition, I have checked all the selected days as well as all the unselected days to make sure our method works.

In addition, even taking the averages over each sea breeze type at each hour is at least questionable. The authors make the implicit assumption that each sea breeze type evolves exactly the same at each hour and therefore taking the average at hour, say, 11 LT is meaningful. But this is not true, differences occur at 11 LT due to the season, due to the position of the sea breeze front, due to the background wind flow, to list a few. Aside from vectors, this is especially troublesome with convergence and divergence fields used to identify the average position of the sea breeze front, because averaging a positive and a zero or negative value at a grid cell, for example, could dilute the signal of the sea breeze location. The authors need to find alternative methods to characterize the statistics of the sea breezes, for example using median values or some pattern recognition techniques.

We understand that each sea breeze case is different from one another. By taking the average of a specific sea breeze type events, the mean development of the sea breeze is revealed (Figures 6-9). The effectiveness of our method is demonstrated as key sea breezes features (e.g., calm zone, coastal jet) appear in the composite figures.

We acknowledge your concern about the differences between individual cases. In this revision, we have conducted additional analysis to examine the variability of simulated sea breeze events. Our results suggest that the temporal development of the calm zone for the pure sea breeze and the positioning of the coastal jet for the corkscrew sea breeze is rather consistent across their identified cases respectively.

[Figure]

To do that, we have defined three regions to quantify the variability of the identified sea breeze cases (as shown in the figure). They are located on land (blue), over the coast (red) and over the ocean (green). The size of region is about 3 % of the entire regional domain. For each sea breeze type, we calculate the standard deviation of WS10 and WD10 from the identified sea breeze events over all three regions from 08 LT to 20 LT, and the results are shown in the tables below.

Table1 : Variability of simulated pure sea breeze cases over land, coast region and ocean

| Standard Deviation of WS10(m/s) for the Identified Pure Sea Breeze Cases | | | | | | | | | | | | |
| --- | --- | --- | --- | --- | --- | --- | --- | --- | --- | --- | --- | --- |
| | 08 LT | 09 LT | 10 LT | 11 LT | 12 LT | 13 LT | 14 LT | 15 LT | 16 LT | 17 LT | 18 LT | 19 LT | 20 LT |
| Inland | 1.4 | 1.3 | 1.2 | 1.1 | 1.2 | 1.3 | 1.4 | 1.3 | 1.1 | 0.9 | 1.0 | 0.8 | 0.8 |
| Coast | 2.3 | 2.0 | 1.8 | 1.6 | 1.5 | 1.4 | 1.3 | 1.4 | 1.4 | 1.4 | 1.3 | 1.2 | 1.4 |
| Ocean | 3.0 | 2.8 | 2.5 | 2.1 | 1.8 | 1.7 | 1.7 | 1.6 | 1.5 | 1.6 | 1.7 | 1.8 | 1.8 |
| Standard Deviation of WD10 (degree) for the Identified Pure Sea Breeze Cases | | | | | | | | | | | | |
| | 08 LT | 09 LT | 10 LT | 11 LT | 12 LT | 13 LT | 14 LT | 15 LT | 16 LT | 17 LT | 18 LT | 19 LT | 20 LT |
| Inland | 119 | 111 | 94 | 77 | 62 | 58 | 58 | 62 | 63 | 63 | 56 | 51 | 55 |
| Coast | 102 | 119 | 114 | 105 | 92 | 78 | 65 | 60 | 55 | 54 | 53 | 62 | 51 |
| Ocean | 118 | 129 | 112 | 108 | 107 | 116 | 115 | 110 | 106 | 97 | 85 | 78 | 76 |

For the pure sea breeze cases (Table1), the variability of WS10 is largest during the morning hours and decreases after that. Overall, the variable of WS10 is greater over the ocean than that on land. As for WD10, the variability is large during the morning hours. Note that, based on our methodology and the shape of the coastline, the pure sea breeze is identified from potential days of three different wind

regimes (Northwesterly, Northly and Westly). Therefore, it is not a surprise that variability of WD10 is large during the morning hour. However, variability of WD10 drastically decreases after the morning hour due to the influence of sea breeze development. Note that the standard deviation of WD10 over the ocean is relatively large until late afternoon. This is mainly due to the development of the calm zone (Figure 6 of the manuscript). After the calm zone moved away from the coast, standard deviation of WD10 reduces significantly (16 LT to 20 LT).

Table2 : Variability of simulated corkscrew sea breeze cases over land, coast region and ocean

| Standard Deviation of WS10(m/s) for the Identified Pure Sea Breeze Cases | | | | | | | | | | | | |
|---|---|---|---|---|---|---|---|---|---|---|---|---|
| | 08 LT | 09 LT | 10 LT | 11 LT | 12 LT | 13 LT | 14 LT | 15 LT | 16 LT | 17 LT | 18 LT | 19 LT | 20 LT |
| Inland | 1.1 | 1.0 | 1.1 | 1.1 | 1.1 | 1.0 | 0.9 | 0.8 | 0.6 | 0.8 | 1.0 | 0.6 | 0.7 |
| Coast | 1.8 | 1.7 | 1.6 | 1.7 | 1.9 | 1.7 | 1.8 | 1.7 | 1.8 | 1.8 | 1.7 | 1.6 | 1.8 |
| Ocean | 2.4 | 2.3 | 2.1 | 1.9 | 1.8 | 1.9 | 2.1 | 2.2 | 2.3 | 2.5 | 2.4 | 2.4 | 2.2 |

| Standard Deviation of WD10 (degree) for the Identified Pure Sea Breeze Cases | | | | | | | | | | | | |
|---|---|---|---|---|---|---|---|---|---|---|---|---|---|
| | 08 LT | 09 LT | 10 LT | 11 LT | 12 LT | 13 LT | 14 LT | 15 LT | 16 LT | 17 LT | 18 LT | 19 LT | 20 LT |
| Inland | 68 | 67 | 58 | 48 | 42 | 43 | 44 | 43 | 43 | 44 | 40 | 46 | 40 |
| Coast | 79 | 80 | 69 | 55 | 45 | 36 | 32 | 35 | 28 | 28 | 28 | 35 | 31 |
| Ocean | 67 | 78 | 85 | 75 | 75 | 78 | 66 | 62 | 60 | 57 | 48 | 36 | 31 |

Table 2 shows the results from the corkscrew sea breezes. In general, the characteristics are similar to that from the pure sea breeze cases. One important aspect is that the small variabilities of WD10 over the coast region during the late afternoon hours. This suggests that the position of the simulated jet core (Figure 7 of the manuscript) over this region is rather stable, which would have significant offshore wind energy implication in terms of wind turbine positioning.

Table3 : Variability of simulated backdoor sea breeze cases over land, coast region and ocean

| Standard Deviation of WS10(m/s) for the Identified Pure Sea Breeze Cases | | | | | | | | | | | | |
|---|---|---|---|---|---|---|---|---|---|---|---|---|---|
| | 08 LT | 09 LT | 10 LT | 11 LT | 12 LT | 13 LT | 14 LT | 15 LT | 16 LT | 17 LT | 18 LT | 19 LT | 20 LT |
| Inland | 0.9 | 1.1 | 1.6 | 1.7 | 1.6 | 1.7 | 1.7 | 1.7 | 1.8 | 1.4 | 0.8 | 0.4 | 0.3 |
| Coast | 2.4 | 2.3 | 2.2 | 2.4 | 2.5 | 2.4 | 2.3 | 2.3 | 2.3 | 2.3 | 2.2 | 2.3 | 2.4 |
| Ocean | 3.5 | 3.7 | 3.6 | 3.5 | 3.4 | 3.0 | 2.9 | 3.0 | 3.0 | 3.0 | 2.9 | 2.8 | 2.7 |

| Standard Deviation of WD10 (degree) for the Identified Pure Sea Breeze Cases | | | | | | | | | | | | |
|---|---|---|---|---|---|---|---|---|---|---|---|---|---|
| | 08 LT | 09 LT | 10 LT | 11 LT | 12 LT | 13 LT | 14 LT | 15 LT | 16 LT | 17 LT | 18 LT | 19 LT | 20 LT |
| Inland | 16 | 17 | 20 | 29 | 34 | 37 | 35 | 31 | 31 | 27 | 27 | 29 | 31 |
| Coast | 13 | 14 | 17 | 18 | 16 | 16 | 17 | 16 | 18 | 22 | 22 | 28 | 26 |
| Ocean | 107 | 94 | 42 | 37 | 58 | 65 | 78 | 62 | 36 | 24 | 17 | 22 | 29 |

Table 3 shows the results from the backdoor sea breezes. Because of low occurrence rate, It has the smallest variability, which also indicates that the development of the individual backdoor sea breeze does not differ much from the mean condition (Figure 8 of the manuscript).

Corresponding texts and tables have been added to the manuscript. Note that we have changed the alignment of three regions in other attempts, such as horizontal and vertical. However, that does not change the table results significantly.

The last major issue is probably just a matter of explaining things better. There must be a sub-region or location of focus of the study, otherwise how can there be one sea breeze type for the entire domain 2? If the sea breeze is affecting New Jersey, it must be east-to-west, but in the northern shores of Long Island is it north-to-south? If it's a pure sea breeze in New York City, could it be corkscrew somewhere else? I suspect that the issue is partly due to Figure 3, which I find very obscure. Where is the land? Is the prevailing wind the geostrophic wind? Where is north and south? Which way is the sea breeze flow?

Response: Thank you for your comment. You are right. We are not detecting sez breeze over the entire domain but only a subregion which is square-shaped area covering the New York metropolitan region (Figure 2b of the manuscript). In this revision, we have reworked Figure 3 to avoid further confusion (see figure below). It shows three idealized scenarios where the arrow indicates geostrophic wind.

[Figure]

Minor issues

1. Why were the 4 quadrants introduced? I don't understand their purpose as they are not used. The text near line 130 talks about "mean … for each individual quadrant", is this an area average over all grid points in the quadrant? Then a

"dominant wind regime" for that day is obtained. What does dominant mean? How many hours out of 24? What if different quadrants had different sea breeze types?

Response: Thank you for your comment. The 4 quadrants are introduced to help detect sea breeze events in our simulation. They are heavily used in our detection method to determine the wind regime and categorize sea breeze type. The dominant wind regime for the region (all 4 quadrants) is determined when at least 2 of 4 quadrants share the same wind regime (NW, W, N, BD and CS). This is step 1 of our detection method and it only examines SLP and wind field at 08 LT for each particular day. The identified days have the potential for sea breeze development and are subjected to step 2 of our detection method. Please see section 2.2 of the manuscript for more details.

2. Table 2: It suggests that 246 days had sea breezes in a year, which seems too many. Again, maybe the sea breeze types are not mutually exclusive, but then I do not understand how the averages are even calculated.

Response: Thank you for your comment. The 246 days identified in Table 2 suggest days with the potential for sea breeze development. This is the results from the Step 1 of our detection method based on wind regime characteristic. These days are later subjected to Step 2 of the detection method and a total of 61 simulated sea breeze events have been identified.

3. 185: here it seems that only 61 days were identified, but Table 2 is not consistent.

Response: Thank you for your comment. Please refers to Minor comment 2 for more details.

4. 205: again, I am confused about the averaging, are you averaging over all 3 types of pure sea breeze here? If so, it seems even more questionable to average over such a broad range of wind directions.

Response: Thank you for your comment. Figure 6 shows the average of all the 28 identified pure sea breeze events. In addition, there is no 3 types of pure sea breeze in our study, but only 3 types of sea breeze (pure, corkscrew and backdoor).

5. Figure 6: even given the fourth major issue above, it seems to me that the only location with a sea breeze here is Long Island.

Response: Thank you for your comment. From 12 to 15 LT, there is also sea breeze occurring over the coast of New Jersey, as evidenced from the surface divergent map.

[Figure]

6. Figure 8: The corkscrew sea breeze seems to be in New Jersey only.

   Response: Thank you for your comments but I think you are referencing to Figure 7. Based on the plot, it does seem like New Jersey is mostly impacted by corkscrew sea breeze. Corresponding sentences has been added to the revised manuscript.

7. Figure 9: because these figures are averages, the dynamic evolution is basically lost. There is no meaningful difference between the fields at 12, 13, and 14 for the pure sea breeze case, for example. Averaging out conv/div, the signal is diluted and the front is less distinguishable.

   Response: Thank you for your comment. The rational of using averaging are provided in the response to the Major comments. Based on Figure 9, the position of the sea breeze front is different for each sea breeze types. In this revision, we change the Figure to show the difference at 09, 13 and 17 LT so that the evolution of the sea breeze front for each sea breeze type can be seen clearly. Corresponding sentences has been changed in the revised manuscript.

8. Line 260: I disagree that "Overall, the results indicate …" There is no evidence that the method works!

   Response: Thank you for your comment. As we mentioned in our response to your Major comment, this is not a validation study. We think our method works as it has successfully identified different sea breeze events as well as the associated sea breeze characteristics from the model simulation.

9. Line 292: which 10-MW turbine?

   Response: Thank you for your comment. We use the power coefficient from IEA 10MW RWT.

10. Line 293: why was this location chosen? Is it where a lease area is proposed? Please explain.

Response: Thank you for your comment. This area is subject for major offshore wind farm construction, and we have mentioned that in the introduction section of the revised manuscript. Please refers to our response to Major comment 1 for more details.

11. Line 319: Your methodology of averaging out everything washes out the details of the timing and evolution of the sea breezes, that is why your results are not consistent with past studies.

Response: Thank you for your comment. The rational of using averaging are provided in the response to the Major comments. This is a modeling study and the first modeling study showing the spatiotemporal evolution of different types of sea breezes whereas the past studies are mostly observational and are for point measurements. Therefore, certain discrepancies between these two types of studies are expected.

---

## Author Comment (AC4)

Review of : WES-2021-109
Detecting and Characterizing Sea Breezes Over the U.S. Northeast Coast with Implication for Offshore Wind Energy
by Xia et al. Description:

The study applies a new two-step classification method for sea breezes using simulations with the WRF regional model. The approach is used with a year of high resolution, 2 km, simulation over the area of New York. The method is used to identify pure, corkscrew, and backdoor breezes, analyze their statistics of occurrence and their impact on energy production.

General comment:

I think that the purpose of the paper is valuable. It is well written in structure and provides relevant results and discussion. I support publication after the minor comments that follow below.

Response: Thank you for your positive views on our paper. We sincerely appreciate the time you spent reviewing this work. In this revision, we have revised the paper substantially based on yours and other reviewer's comment. The key changes are

- The title of the paper has changed to *Detecting and Characterizing **Simulated** Sea Breezes Over the U.S. Northeast Coast with Implication for Offshore Wind Energy*.
- An additional analysis has been conducted to examine the variability of individual sea breeze cases.

Specific comments

SC1 Section 2.1. Experiment design. A ratio of 3:1 is most often used in the design of the model domains. I suggest the authors include some comments about the use of a 5:1 ratio. Some arguments on the final resolution selected would also be welcome. For instance, a downscaling enhancing the resolution from the ca. 27 Km of ERA5 to 9 and 3 km would be another possibility, even down to 1 km. Also, some arguments about the selection of parameterizations would be good, specifically the use of microphysics. Overall, it would be good to include some rationale about the model configuration selected.

Response: Thank you for your comment. In general, the WRF developers recommend using odd ratio – typically 3:1 or 5:1. The 5:1 ratio has been suggested years ago. The same WRF configuration has been used in a different study (Pronk et al. 2021). Corresponding reference has been added to the revised manuscript.

Reference:

Pronk, V., Bodini, N., Optis, M., Lundquist, J. K., Moriarty, P., Draxl, C., Purkayastha, A., and Young, E.: Can Reanalysis Products Outperform Mesoscale Numerical Weather Prediction Models in Modeling the Wind Resource in Simple Terrain?, Wind Energ. Sci. Discuss. https://doi.org/10.5194/wes-2021-97, in review, 2021.

SC2 This is not a model evaluation paper. However, it would be an asset that figures 6 and 7 of the composite averages would also show some observed values. This would allow for assessing consistency with observations. One single station would allow for verifying the wind rotation.

Response: Thank you for your comment. As you can tell, this paper is not about model evaluation or model validation. This is also the reason why we change the title to better reflect our purpose.

Validating with observations is certainly the second step of this work but is too much to cover in this paper as there are over 60 identified simulated sea breeze cases. Such analysis would be better fitted for a case study where we focus on comparing not just the wind field but also the other metrological aspects (e.g., temperature, vertical structure) between the simulated and observed sea breeze.

SC3 I think it would be important to provide some information on the representativeness of the average maps in figs 6 and 7. It would be useful to provide some evaluation of the variability of each hour within the composite, for instance, with the map of local variances. Alternatively, a wind rose for a point in the center of the domain could be shown using all the days of the composite.

Response: Thank you for your comment. In this revision, we have conducted additional analysis to examine the variability of simulated sea breeze events to address your concern. Our results suggest that the temporal development of the calm zone for the pure sea breeze and the positioning of the coastal jet for the corkscrew sea breeze is rather consistent across their identified cases respectively.

[Figure]

To do that, we have defined three regions to quantify the variability of the identified sea breeze cases (as shown in the figure). They are located on land (blue), over the coast (red) and over the ocean (green). The size of region is about 3 % of the entire regional domain. For each sea breeze type, we calculate the standard deviation of WS10 and WD10 from the identified sea breeze events over all three regions from 08 LT to 20 LT, and the results are shown in the tables below.

Table1 : Variability of simulated pure sea breeze cases over land, coast region and ocean

| Standard Deviation of WS10(m/s) for the Identified Pure Sea Breeze Cases | | | | | | | | | | | | |
| | 08 LT | 09 LT | 10 LT | 11 LT | 12 LT | 13 LT | 14 LT | 15 LT | 16 LT | 17 LT | 18 LT | 19 LT | 20 LT |
|---|---|---|---|---|---|---|---|---|---|---|---|---|---|
| Inland | 1.4 | 1.3 | 1.2 | 1.1 | 1.2 | 1.3 | 1.4 | 1.3 | 1.1 | 0.9 | 1.0 | 0.8 | 0.8 |
| Coast | 2.3 | 2.0 | 1.8 | 1.6 | 1.5 | 1.4 | 1.3 | 1.4 | 1.4 | 1.4 | 1.3 | 1.2 | 1.4 |
| Ocean | 3.0 | 2.8 | 2.5 | 2.1 | 1.8 | 1.7 | 1.7 | 1.6 | 1.5 | 1.6 | 1.7 | 1.8 | 1.8 |
| Standard Deviation of WD10 (degree) for the Identified Pure Sea Breeze Cases | | | | | | | | | | | | |
| | 08 LT | 09 LT | 10 LT | 11 LT | 12 LT | 13 LT | 14 LT | 15 LT | 16 LT | 17 LT | 18 LT | 19 LT | 20 LT |
| Inland | 119 | 111 | 94 | 77 | 62 | 58 | 58 | 62 | 63 | 63 | 56 | 51 | 55 |
| Coast | 102 | 119 | 114 | 105 | 92 | 78 | 65 | 60 | 55 | 54 | 53 | 62 | 51 |
| Ocean | 118 | 129 | 112 | 108 | 107 | 116 | 115 | 110 | 106 | 97 | 85 | 78 | 76 |

For the pure sea breeze cases (Table1), the variability of WS10 is largest during the morning hours and decreases after that. Overall, the variable of WS10 is greater over the ocean than that on land. As for WD10, the variability is large during the morning hours. Note that, based on our methodology and the shape of the coastline, the pure sea breeze is identified from potential days of three different wind regimes (Northwesterly, Northly and Westly). Therefore, it is not a surprise that variability of WD10 is large during the morning hour. However, variability of WD10 drastically decreases after the morning hour due to the influence of sea breeze development. Note that the standard deviation of WD10 over the ocean is relatively large until late afternoon. This is mainly due to the development of the calm zone (Figure 6 of the manuscript). After the calm zone moved away from the coast, standard deviation of WD10 reduces significantly (16 LT to 20 LT).

Table2 : Variability of simulated corkscrew sea breeze cases over land, coast region and ocean

| Standard Deviation of WS10(m/s) for the Identified Pure Sea Breeze Cases | | | | | | | | | | | | |
| | 08 LT | 09 LT | 10 LT | 11 LT | 12 LT | 13 LT | 14 LT | 15 LT | 16 LT | 17 LT | 18 LT | 19 LT | 20 LT |
|---|---|---|---|---|---|---|---|---|---|---|---|---|---|
| Inland | 1.1 | 1.0 | 1.1 | 1.1 | 1.1 | 1.0 | 0.9 | 0.8 | 0.6 | 0.8 | 1.0 | 0.6 | 0.7 |
| Coast | 1.8 | 1.7 | 1.6 | 1.7 | 1.9 | 1.7 | 1.8 | 1.7 | 1.8 | 1.8 | 1.7 | 1.6 | 1.8 |
| Ocean | 2.4 | 2.3 | 2.1 | 1.9 | 1.8 | 1.9 | 2.1 | 2.2 | 2.3 | 2.5 | 2.4 | 2.4 | 2.2 |

| Standard Deviation of WD10 (degree) for the Identified Pure Sea Breeze Cases | | | | | | | | | | | | |
| | 08 LT | 09 LT | 10 LT | 11 LT | 12 LT | 13 LT | 14 LT | 15 LT | 16 LT | 17 LT | 18 LT | 19 LT | 20 LT |
|---|---|---|---|---|---|---|---|---|---|---|---|---|---|
| Inland | 68 | 67 | 58 | 48 | 42 | 43 | 44 | 43 | 43 | 44 | 40 | 46 | 40 |
| Coast | 79 | 80 | 69 | 55 | 45 | 36 | 32 | 35 | 28 | 28 | 28 | 35 | 31 |
| Ocean | 67 | 78 | 85 | 75 | 75 | 78 | 66 | 62 | 60 | 57 | 48 | 36 | 31 |

Table 2 shows the results from the corkscrew sea breezes. In general, the characteristics are similar to that from the pure sea breeze cases. One important aspect is that the small variabilities of WD10 over the coast region during the late afternoon hours. This suggests that the position of the simulated jet core (Figure 7 of the manuscript) over this region is rather stable, which would have significant offshore wind energy implication in terms of wind turbine positioning.

Table3 : Variability of simulated backdoor sea breeze cases over land, coast region and ocean

| Standard Deviation of WS10(m/s) for the Identified Pure Sea Breeze Cases | | | | | | | | | | | | |
| | 08 LT | 09 LT | 10 LT | 11 LT | 12 LT | 13 LT | 14 LT | 15 LT | 16 LT | 17 LT | 18 LT | 19 LT | 20 LT |
|---|---|---|---|---|---|---|---|---|---|---|---|---|---|
| Inland | 0.9 | 1.1 | 1.6 | 1.7 | 1.6 | 1.7 | 1.7 | 1.7 | 1.8 | 1.4 | 0.8 | 0.4 | 0.3 |
| Coast | 2.4 | 2.3 | 2.2 | 2.4 | 2.5 | 2.4 | 2.3 | 2.3 | 2.3 | 2.3 | 2.2 | 2.3 | 2.4 |
| Ocean | 3.5 | 3.7 | 3.6 | 3.5 | 3.4 | 3.0 | 2.9 | 3.0 | 3.0 | 3.0 | 2.9 | 2.8 | 2.7 |

| Standard Deviation of WD10 (degree) for the Identified Pure Sea Breeze Cases | | | | | | | | | | | | |
| | 08 LT | 09 LT | 10 LT | 11 LT | 12 LT | 13 LT | 14 LT | 15 LT | 16 LT | 17 LT | 18 LT | 19 LT | 20 LT |
|---|---|---|---|---|---|---|---|---|---|---|---|---|---|
| Inland | 16 | 17 | 20 | 29 | 34 | 37 | 35 | 31 | 31 | 27 | 27 | 29 | 31 |
| Coast | 13 | 14 | 17 | 18 | 16 | 16 | 17 | 16 | 18 | 22 | 22 | 28 | 26 |
| Ocean | 107 | 94 | 42 | 37 | 58 | 65 | 78 | 62 | 36 | 24 | 17 | 22 | 29 |

Table 3 shows the result*s* from the backdoor sea breezes. Because of low occurrence rate, It has the smallest variability, which also indicates that the development of the individual backdoor sea breeze does not differ much from the mean condition (Figure 8 of the manuscript).

Corresponding texts and tables have been added to the manuscript. Note that we have changed the alignment of three regions in other attempts, such as horizontal and vertical. However, that does not change the table results significantly.

SC4 Consider including in Table 2 the numbers of sea breeze events, and also unclassified events.

Response: Thank you for your comment. The unidentified case has been added to Table 2.

SC5 I like the discussion in Section 4. I think that some more comments arguing about the potential sensitivity of the results to changes in the WRF configuration or changes in the statistical approach to detect breezes would enrich the Section. The discussion includes an assessment of the sensitivity of the current approach to changes in the parameters used. What would arguably be the impact of other methods?. In any case, the fundamental conclusion that identifying different types of breezes is relevant for wind energy forecasts is not expected to change.

Response: Thank you for your comment: The following sentences have been added to enrich the discussion.

"*Other factors such as changes in the WRF configuration, statistical approach and targeted region, could have potential sensitivity to the overall number as well as the seasonal distribution for each type of sea breeze. Nevertheless, the importance of identifying the correct type of sea breeze for wind energy forecast would still be significant and serve as a high-priority research topic, especially for offshore wind energy.*

Minor comments

MC1 Abstract, lines 20-22: 'From the wind energy perspective, the power production associated with a 10 megawatts offshore wind turbine would produce approximately 3 to 4 times more electrical power during a corkscrew sea breeze event than the other two types of sea breezes' The sentence reads strangely to me. Alternative '… would be approximately 3 to 4 times larger during…'

Response: Thank you for your comment. Corresponding changes have been made.

MC2 Lines 75: '(Hersbach et al., 2020)'

Response: Thank you for your comment. Corresponding reference has been added.

MC3 In general the characters of the figures are small. Making them larger would produce a better reading experience if the manuscript is printed

Response: Thank you for your comment. Corresponding changes have been made.

MC4 In general the characters of the figures are small. Making them larger would produce a better reading experience if the manuscript is printed

Response: Thank you for your comment. Corresponding changes have been made.

MC5 '…between 08 to 20 local time (LT),…' I suggest using LST, local standard time, that here would be Eastern time. The chances that LT gets confused with saving time are low but I think it would be more appropriate

Response: Thank you for your comment. In general, LST stands for "Land Surface Temperature". So we keep the LT as acronym to avoid confusion.

MC6 '… WD10 for each individual quadrantS …'

Response: Thank you for your comment. Corresponding changes have been made.

MC7 Figure 3 caption. Alternative: '…prevailing wind where SLP conditions favor the development of: a) pure sea breeze; b) backdoor sea breeze; c) corkscrew sea breeze. '

Response: Thank you for your comment. Corresponding changes have been made.